# FL-Sailer: Efficient and Privacy-Preserving Federated Learning for Scalable Single-Cell Epigenetic Data Analysis via Adaptive Sampling

**Guangyi Zhang**[1]**, Yi Dai**[1,†]**, Yiyun He**[2,†]**, Junhao Liu**[1]

[1]Department of Computer Science, University of California, Irvine
[2]Department of Mathematics, University of California, San Diego
[†]Equal contribution in alphabetical order.

**Reviewed on OpenReview:** https://openreview.net/forum?id=2vNebz5r4b

## Abstract

Single-cell ATAC-seq (scATAC-seq) enables high-resolution mapping of chromatin accessibility, yet privacy regulations and data size constraints hinder multi-institutional sharing. Federated learning (FL) offers a privacy-preserving alternative, but faces three fundamental barriers in scATAC-seq analysis: ultra-high dimensionality, extreme sparsity, and severe cross-institutional heterogeneity. We propose FL-Sailer, the first FL framework designed for scATAC-seq data. FL-Sailer integrates two key innovations: (i) adaptive leverage score sampling, which selects biologically interpretable features while reducing dimensionality by 80%, and (ii) an invariant VAE architecture, which disentangles biological signals from technical confounders via mutual information minimization. We provide a convergence guarantee, showing that FL-Sailer converges to an approximate solution of the original high-dimensional problem with bounded error. Extensive experiments on synthetic and real epigenomic datasets demonstrate that FL-Sailer not only enables previously infeasible multi-institutional collaborations but also surpasses centralized methods by leveraging adaptive sampling as an implicit regularizer to suppress technical noise. Our work establishes that federated learning, when tailored to domain-specific challenges, can become a superior paradigm for collaborative epigenomic research.

## 1 Introduction

Recent advances in single-cell epigenomic sequencing have revolutionized genomics by enabling the simultaneous molecular profiling of millions of cells (Fang et al., 2021; Ashuach et al., 2022; Chen et al., 2021a; Xu et al., 2022). These technologies provide unprecedented resolution for characterizing cellular heterogeneity in complex tissues, advancing the foundational goals of precision medicine. Concurrently, transparent data-sharing initiatives support the integration of population-scale single-cell atlases, facilitating the dissection of disease mechanisms across diverse patient cohorts and accelerating the development of personalized diagnostic and therapeutic strategies. However, two critical challenges still impede our progress. First, genomic datasets often contain sensitive clinical information, compelling most data to remain siloed across institutions under stringent privacy regulations (Bonomi et al., 2020; Clayton et al., 2019; Shabani & Borry, 2018; Naveed et al., 2015), which severely restricts access. Second, the immense size of single-cell data—often reaching tens of terabytes—imposes prohibitive bandwidth and storage demands, rendering cross-site transfer of high-dimensional epigenetic profiles such as scATAC-seq (Cusanovich et al., 2015; Hwang et al., 2025; Emani et al., 2024) infeasible. These barriers underscore an urgent need for computational frameworks that ensure privacy protection and efficient data integration in single-cell studies (Bonomi et al., 2020; Pfitzner et al., 2021).

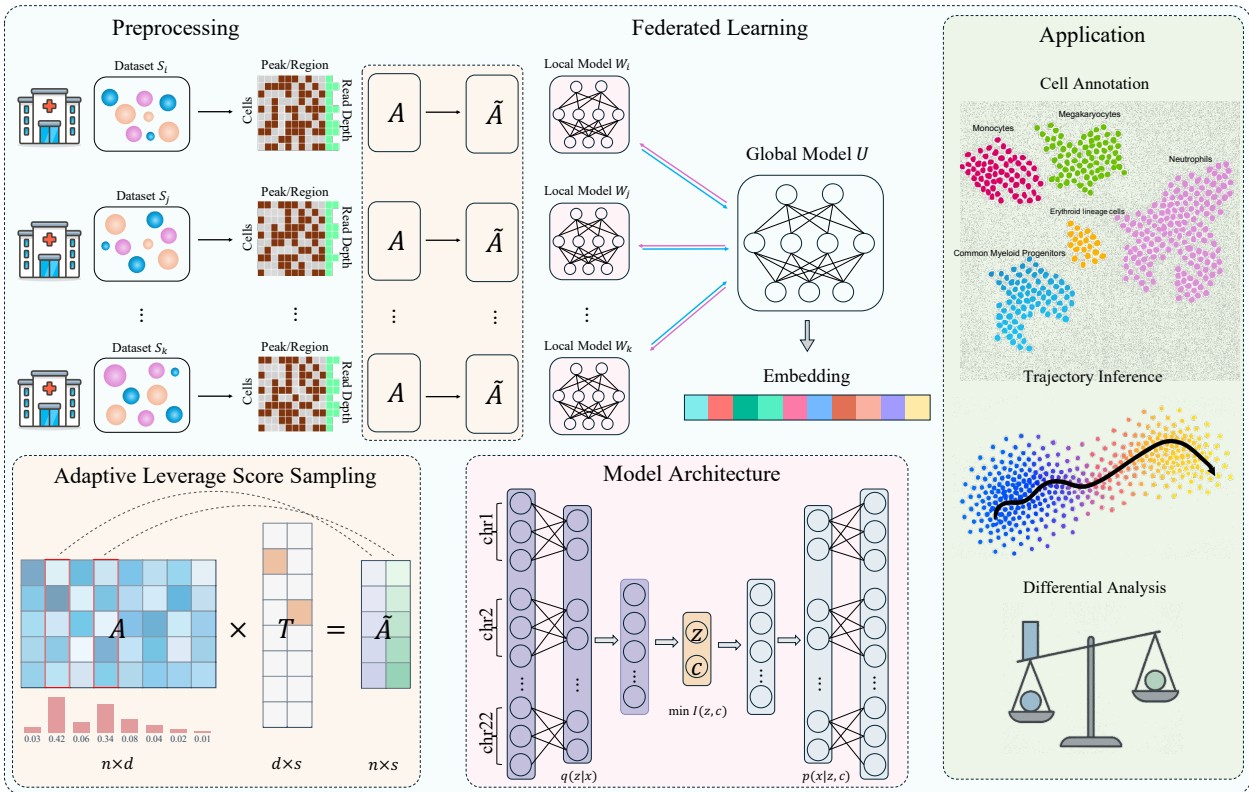

Figure 1: **FL-Sailer: Federated Learning for Single-Cell Epigenomics.** FL-Sailer makes federated learning feasible for million-feature genomic data by jointly addressing dimensionality (adaptive leverage score sampling: $d \to s$, 80% reduction) and heterogeneity (invariant VAE: $I(z, c)$ minimization), transforming a computationally impossible problem into a practical solution with superior performance.
**Pipeline:** (1) Clients subsample local $n \times d$ matrices to $n \times s$ via leverage scores; (2) Federated training of chromosome-block VAE disentangles biological signals $z$ from technical confounders $c$; (3) Unified latent space enables multi-institutional downstream discovery.

Federated learning (FL) methods have been proposed to address these challenges by training models collaboratively without moving the raw data from their original locations. Instead of centralizing datasets, the approach distributes the learning process by sending a global model to each local site for training on its private data. Only the model updates, rather than the data itself, are transmitted back to be aggregated into an improved global model. This method not only preserves data privacy and compliance but also efficiently leverages large, distributed datasets by parallelizing the computational workload across multiple institutions. However, while FL has proven successful in non-biological domains (McMahan et al., 2017; Reddi et al., 2020; Li et al., 2021; 2022), standard FL fails on collaborative single cell epigenetic data due to **three compounding barriers**: ultra-high dimensionality ($10^5$–$10^7$ features) requiring prohibitive gigabyte-scale communication, extreme sparsity ($\sim$95%) that obscures biological signals, and severe cross-institutional heterogeneity from technical and biological confounders. Therefore, how to adapt FL algorithms to the single-cell field is still a unresolved question.

To address this, we present FL-Sailer, a federated learning framework engineered to overcome the barriers to multi-institutional single-cell epigenomic analysis. FL-Sailer avoids centralizing sensitive raw data and aligns with common privacy constraints in multi-institutional genomic studies. It simultaneously addresses the critical challenges of communication efficiency and model performance, preventing the degradation that typically plagues distributed learning on heterogeneous genomic datasets. This approach enables robust, scalable, and privacy-conscious collaboration, facilitating studies that were previously infeasible. The major contributions of this work are:

- **Adaptive leverage score sampling that preserves biological interpretability.** Our sampling module directly selects genomic regions, yielding features with direct biological interpretability while

reducing communication cost. For extreme-dimensional datasets (up to 423K features), this achieves 80% dimension reduction and orders-of-magnitude communication savings. More importantly, it actually improves clustering performance over standard methods by focusing computation on high-signal regulatory regions.

- **Invariant representation learning for robust handling of batch effects.** FL-Sailer's VAE architecture explicitly disentangles biological signals from technical confounders through mutual information minimization. This directly addresses severe batch effects common in multi-institutional data, where sequencing depth varies 50-fold and technical variations correlate with institutions. Our approach ensures meaningful biological discovery across datasets with different experimental conditions.

- **First convergence theory for federated learning with aggressive feature sampling.** We establish that FL-Sailer maintains linear convergence under extreme dimension reduction, with approximation error $O(\epsilon \cdot L(U^*))$ vanishing as $\epsilon = O(\sqrt{r/(\rho d)}) \to 0$ for high-dimensional data (Theorem 5.1). This extends non-convex FL theory to the previously unanalyzed regime of aggressive sampling, proving that substantial dimension reduction is both necessary for privacy-preserving genomic analysis and theoretically guaranteed.

- **Systematic benchmarking on diverse simulated and real datasets.** We performed extensive benchmarking to evaluate our framework. FL-Sailer avoids centralizing raw profiles and supports privacy-conscious multi-institutional analysis, while robustly outperforming conventional centralized models. These results establish FL, when properly engineered for domain-specific constraints, as a superior paradigm for collaborative epigenomic studies.

We release FL-Sailer as a scalable tool for processing millions of cells, enabling privacy-preserving multi-institutional epigenomic studies. The remainder of this paper is organized as follows. Section 2 reviews related work, and Section 3 formulates the federated single-cell analysis problem. Sections 4 and 5 present our theoretical framework: Section 4 analyzes adaptive sampling and invariant learning, while Section 5 establishes end-to-end convergence guarantees for FL-Sailer. Section 6 evaluates FL-Sailer on simulated and real-world datasets, where it enables otherwise infeasible analyses and often outperforms centralized methods. We conclude with its implications for collaborative epigenomic research.

## 2 Related Work

### 2.1 The Centralized Paradigm: Established Tools for Single-Cell Epigenomics

Single-cell epigenomic sequencing, particularly scATAC-seq, has revolutionized the study of gene regulation by profiling chromatin accessibility at unprecedented resolution (Buenrostro et al., 2015; Cusanovich et al., 2015). To analyze these high-dimensional datasets, computational methods have evolved from linear statistical baselines (e.g., LSI, chromVAR) to deep generative models like SCALE (Xiong et al., 2019) and PeakVI, which capture nonlinear latent structures from sparse counts. Among these, SAILER (Cao et al., 2021) represents a significant milestone, employing a chromosome-block VAE with invariant representation learning to explicitly disentangle biological signals from technical confounders. Despite its effectiveness, SAILER fundamentally relies on pooling data into a single repository. As dataset scales grow to millions of cells and privacy regulations (e.g., GDPR, HIPAA) intensify, the centralized training paradigm becomes increasingly untenable, necessitating a shift toward decentralized frameworks.

### 2.2 The Federated Frontier: Decentralized Learning for Biomedical Data Analysis

FL has emerged as the standard for collaborative model training without data sharing (McMahan et al., 2017). Its utility is well-documented in biomedical domains ranging from genome-wide association studies (GWAS) (Cho et al., 2025) to rare cancer detection (Pati et al., 2022) and multi-omics integration (Zhou et al., 2024). Recent frameworks such as scPrivacy (Chen et al., 2023) and scFed (Wang et al., 2024) demonstrate the feasibility of federated learning for scRNA-seq analysis. We view them as important related works, but they were developed for a different operating regime from the one studied here. In particular, scRNA-seq inputs are typically on the order of $10^4$ genes, whereas the scATAC-seq datasets considered in this work

contain 108K–423K genomic regions with 93%–99% sparsity. In our setting, this order-of-magnitude increase materially changes the communication and optimization regime: model size and per-round communication scale with the number of retained peaks, making naive federated training prohibitively expensive without sampling.

### 2.3 Methodological Gaps: Dimensionality, Sparsity, and Theory

The direct adaptation of FL to scATAC-seq is hindered by three fundamental gaps in existing literature:

**The Communication-Computation Gap.** Standard FL algorithms like FedAvg require $O(d)$ communication per round. With scATAC-seq features, this cost is prohibitive. While dimensionality reduction techniques such as leverage score sampling (Drineas et al., 2012) and CUR decompositions (Boutsidis & Woodruff, 2014) are well-established, they are inherently designed for *centralized* matrices. There is currently no efficient, privacy-preserving mechanism to perform importance sampling on distributed, high-dimensional genomic data without first centralizing it.

**The Heterogeneity-Sparsity Gap.** scATAC-seq data are characterized by extreme sparsity ($\sim 95\%$ zeros) and severe cross-institutional batch effects (Stuart et al., 2019). In this regime, local stochastic gradients exhibit high variance and misalignment. Standard aggregation strategies (e.g., weighted averaging in FedAvg) fail to correct for systematic technical confounders across clients, leading to model divergence or the learning of spurious batch-specific features (Karimireddy et al., 2020).

**The Theoretical Gap.** Although recent works have extended FL convergence theory to non-convex settings under relaxed assumptions (Li et al., 2022), these analyses presume that full-gradient transmission is feasible. They do not account for the aggressive feature selection required for genomic scalability. To our knowledge, no prior work provides convergence guarantees for FL that simultaneously addresses ultra-high dimensionality, sparsity, and non-IID heterogeneity—a theoretical void this work aims to fill.

## 3 Overview and Problem Setup

We introduce FL-Sailer, a federated learning framework that facilitates privacy-preserving, collaborative analysis of ultra-high-dimensional single-cell epigenomic data. As illustrated in Figure 1, the FL-Sailer pipeline is specifically designed to overcome the fundamental challenge of computational feasibility in FL for genomic data comprising millions of features, while simultaneously ensuring the preservation of critical biological signals.

### 3.1 Problem Formulation and Notation

We consider $N$ institutions, each holding private scATAC-seq data represented as binary matrices $A_i \in \{0,1\}^{n_i \times d}$, where $d \in [10^5, 10^7]$ denotes the ultra-high dimensionality of genomic regions. The conceptual stacked matrix $A = [A_1; \ldots; A_N]$ encapsulates all distributed data, with rows representing cells and columns representing genomic features. Each cell on client $i$ is characterized by a pair $(x_{ij}, c_{ij})$, where $x_{i,j} := A_i[j,:] \in \{0,1\}^d$ represents the chromatin accessibility profile, and $c_{i,j} \in \mathcal{C}$ denotes technical confounders (e.g., sequencing depth, batch identifier, or institution-specific effects).

Let $[n] = \{1, 2, \ldots, n\}$. For vectors, $\|\cdot\|_2$ denotes the $\ell_2$ norm and $\|\cdot\|_0$ the number of non-zero entries. For matrices, $\|\cdot\|$ denotes the spectral norm and $(\cdot)^\dagger$ the Moore-Penrose pseudoinverse. Throughout this paper, $W$ denotes model parameters in general mathematical statements, while $U$ specifically represents global model parameters in federated algorithms. Tilde notation (e.g., $\tilde{U}$) indicates quantities in the dimensionally-reduced space after feature sampling, with $T$ denoting the sampling transformation matrix.

### 3.2 Centralized task and objective

We first formalize the learning task in the centralized setting, where all scATAC-seq data are accessible in a single matrix $A \in \{0,1\}^{n \times d}$. Our goal is to learn an encoder $q_\theta(z|x)$ that maps a cell's high-dimensional accessibility profile $x$ to a low-dimensional, biologically meaningful latent representation $z$, while being

invariant to technical confounders $c$ (e.g., sequencing batch or depth). A conditional decoder $p_\phi(x|z,c)$ is simultaneously learned to reconstruct the input. This is achieved by minimizing the following objective, adapted from the SAILER framework (Cao et al., 2021):

$$\mathcal{L}(\theta, \phi) = \mathcal{L}_{\text{prior}} + \lambda \mathcal{L}_{\text{marginal}} + (1 + \lambda)\mathcal{L}_{\text{recon}}$$

where the marginal term upper-bounds $I(z;c)$ and the reconstruction term uses a likelihood suitable for sparse/binary counts. At inference, we use the posterior mean of $q_\theta(z|x)$ as the cell embedding; for imputation we decode with $c$ fixed to reference values.

### 3.3 Federated task and assumptions

While the centralized objective in Section 3.2 provides a foundational formulation, it is predicated on direct access to the entire dataset $A$, which is infeasible under the data privacy requirements of multi-institutional collaborations.

**Problem 3.1** (Federated scATAC-seq Analysis). *Our goal is to learn a shared model $U$ that minimizes a global objective function, which is the average of the local objectives from all participating clients:*

$$\min_U \mathcal{L}(U) = \sum_{i=1}^{N} \frac{n_i}{n} \mathcal{L}_i(U), \tag{1}$$

*where $\mathcal{L}_i(U)$ is the local objective function for client $i$, calculated over its private dataset $D_i$. The learning process is subject to three critical constraints, privacy, communication efficiency and robustness.*

Our privacy scope is limited to the standard cross-silo setting in which raw profiles remain local; we do not claim formal differential privacy or attack-specific guarantees in this work.

## 4 Enabling Federated Learning for High-Dimensional Single-Cell Data: Theoretical Foundations

**Overview.** This section develops the theoretical ingredients that make federated single-cell epigenomic analysis both feasible and robust. In Section 4.1 we show that adaptive leverage-score sampling yields a near-isometric subspace embedding for ultra-high-dimensional scATAC-seq matrices, reducing per-round communication from $O(d)$ to $O(s)$ with $s = \Theta(r \log(r/\delta)/\varepsilon^2)$ while preserving optimization geometry and biological structure. In Section 4.2, we adopt an adapted invariant representation for VAE (Cao et al., 2021) and restate the variational bound and resulting objective under our federated, communication-constrained setting. These components set up the end-to-end convergence analysis in Section 5.

### 4.1 Adaptive Feature Selection for Communication Efficiency

To make FL computationally feasible for scATAC-seq with millions of features, we introduce an efficient sampling strategy based on leverage scores that preserves essential biological structure while dramatically reducing communication costs.

**Lemma 4.1** (Feature Leverage Score Sampling). *Let $A \in \mathbb{R}^{n \times d}$ have $\text{rank}(A) = r \leq \min\{n, d\}$. For each $j \in [d]$, define the column leverage score as*

$$\ell_j^{\text{col}} := a_{:,j}^\top \left(A A^\top\right)^\dagger a_{:,j} \geq 0,$$

*where $a_{:,j} \in \mathbb{R}^n$ is the $j$-th column of $A$. Note that $\sum_{j=1}^{d} \ell_j^{\text{col}} = r$. Define the sampling probabilities $p_j = \ell_j^{\text{col}}/r$.*

*Let $s$ be the number of columns to sample. We select a subset of indices $\mathcal{S} = \{j_1, \ldots, j_s\} \subset [d]$ via weighted sampling without replacement, where the sampling probabilities are proportional to $\{p_j\}_{j=1}^{d}$. Let $T \in \mathbb{R}^{d \times s}$ be the sampling and rescaling matrix. For each $m \in \{1, \ldots, s\}$, the $m$-th column of $T$ corresponds to the index*

$j_m \in \mathcal{S}$. *Specifically, $T$ is defined by the non-zero entries $T_{j_m,m} = 1/\sqrt{s \cdot p_{j_m}}$, with all other entries set to zero. The sketched matrix is then constructed as $\tilde{A} = AT \in \mathbb{R}^{n \times s}$.*

*For any $\epsilon \in (0,1)$ and $\delta \in (0,1/2)$, if we choose $s = \Theta(r \log(r/\delta)/\epsilon^2)$, then with probability $1 - \delta$, the following holds:*

$$(1 - \epsilon)\, AA^\top \;\preceq\; \tilde{A}\tilde{A}^\top \;\preceq\; (1 + \epsilon)\, AA^\top.$$

*In other words, with probability $1 - \delta$, for all $y \in \mathbb{R}^n$,*

$$(1 - \epsilon)\, \|A^\top y\|_2^2 \;\leq\; \|(AT)^\top y\|_2^2 \;\leq\; (1 + \epsilon)\, \|A^\top y\|_2^2.$$

**Remark 4.2.** *While classical leverage score sampling typically employs sampling with replacement to guarantee spectral approximation (Drineas et al., 2012; Boutsidis & Woodruff, 2014), we adopt a sampling without replacement strategy. In the context of scATAC-seq, features correspond to specific physical genomic coordinates. Selecting the same genomic region multiple times provides redundant biological information and hinders the interpretability of the selected marker set. By sampling without replacement, we ensure a diverse coverage of distinct regulatory elements. Theoretically, this modification is justified by (Gross & Nesme, 2010, Theorem 1), who established that the Matrix Hoeffding inequalities underpinning our convergence bounds remain valid under the sampling without replacement setting.*

Using this foundational lemma, we now establish our main result on communication efficiency:

**Theorem 4.3** (Communication Efficiency via Feature Leverage Score Sampling)**.** *Let $A \in \mathbb{R}^{n \times d}$ represent scATAC-seq data with $\mathrm{rank}(A) = r \leq \min\{n, d\}$. Following the leverage score sampling framework (Lemma 4.1), we construct a sketched matrix $\tilde{A} = AT \in \mathbb{R}^{n \times s}$, s.t. the communication cost is reduced from $O(d)$ to $O(s)$ in the federated setting, achieving a compression ratio of $O(d/s) = O(d\epsilon^2/(r\log(r/\delta)))$.*

Crucially, the leverage score sampling preserves both the optimization geometry and gradient information in the sampled subspace. Specifically, the solution in the sampled space provides an approximation to the original high-dimensional problem, with bounded reconstruction error and preserved gradient norms (see Appendix B.1 for detailed characterization).

**Biological structure is preserved.** Our leverage score sampling maintains key biological signals while reducing dimensionality. Formally, the sampling preserves (i) cell-type separability, (ii) clustering structure, and (iii) provides theoretical guarantees for selecting biologically informative genomic regions. We defer the formal statement and proof to Appendix B.2. This ensures that restricting optimization and communication to the sampled subspace does not destroy biologically meaningful geometry.

## 4.2 Invariant Representation Learning with Communication Constraints

Having addressed communication efficiency, we now tackle client heterogeneity. We adapt an invariant VAE architecture (Cao et al., 2021) to learn representations robust to technical confounders $c$ (e.g., sequencing depth, batch effects) while preserving biological signals.

Our objective minimizes the mutual information $I(z; c)$ between latent representations $z$ and confounders $c$, ensuring $z$ encodes only technically invariant information. By minimizing a tractable variational upper bound (Appendix C.1), we obtain the federated optimization objective:

**Corollary 4.4** (Optimization Objective, (Cao et al., 2021))**.** *The complete loss function for invariant representation learning, with parameters $W = (\theta, \phi)$, can be expressed as a sum of three key components:*

$$\mathcal{L}(W) = \mathcal{L}_{prior}(W) + \lambda \mathcal{L}_{marginal}(W) + (1 + \lambda)\mathcal{L}_{recon}(W) \tag{2}$$

*where the components are defined as:*

- *Prior KL Divergence: $\mathcal{L}_{prior}(W) = \mathbb{E}_{x \sim q(x)}[KL[q_\theta(z|x)\|p(z)]]$ enforces that posterior distributions remain close to the prior $p(z)$, preventing latent collapse.*

- *Marginal KL (Invariance) Term: $\mathcal{L}_{marginal}(W) = \mathbb{E}_{x \sim q(x)}[KL[q_\theta(z|x)\|q_\theta(z)]]$ minimizes the mutual information $I(z;x)$, which serves as a variational upper bound for $I(z;c)$ to suppress technical confounders via information bottleneck regularization.*

- *Reconstruction Loss: $\mathcal{L}_{recon}(W) = -\mathbb{E}_{x,c}[\mathbb{E}_{z \sim q_\theta(z|x)}[\log p_\phi(x|z,c)]]$ ensures reconstruction fidelity through conditional decoding, with coefficient $(1 + \lambda)$ balancing invariance and reconstruction quality.*

This objective facilitates federated training by allowing clients to learn a unified biological representation $z$, resilient to local technical variations, thus enabling effective model aggregation. Full derivations and regularity assumptions are deferred to Appendices C.1 and C.2.

---

**Algorithm 1** FL-Sailer: Federated Learning with Column Leverage Score Sampling

---

**Require:** Client datasets $\{\mathcal{D}_i\}_{i=1}^N$ with matrices $A_i \in \mathbb{R}^{n_i \times d}$; rounds $R$; sampling rate $\rho$; sketch size $s_k$
**Ensure:** Global model $U^R$

**Phase 1: Federated Dynamic Feature Selection**
1: **for all** $i \in [N]$ **in parallel do**
2:     Generate $\Omega_i \sim \mathcal{N}(0,1)^{s_k \times n_i}$
3:     $B_i \leftarrow \Omega_i A_i \in \mathbb{R}^{s_k \times d}$
4:     Compute Thin QR: $B_i^\top = Q_i R_i$        $\triangleright Q_i \in \mathbb{R}^{d \times s_k}$
5:     $\hat{\ell}_i^{\text{col}}[j] \leftarrow \|(Q_i)_{j,:}\|_2^2, \ \forall j \in [d]$
6:     Send $(\hat{\ell}_i^{\text{col}}, n_i)$ to server
7: **end for**
    **Server: Aggregate and sample**
8: $n \leftarrow \sum_{i=1}^N n_i$
9: $\hat{\ell}^{\text{global}}[j] \leftarrow \sum_{i=1}^N \frac{n_i}{n} \hat{\ell}_i^{\text{col}}[j], \ \forall j$
10: $p_j \leftarrow \hat{\ell}^{\text{global}}[j] / \sum_{j'} \hat{\ell}^{\text{global}}[j']$
11: $s \leftarrow \lfloor \rho d \rfloor$
12: Sample $\mathcal{S} \subset [d]$ of size $s$ using $p$ without replacement
13: Broadcast $\mathcal{S}$ to all clients

**Phase 2: Subspace Federated Training**
1: **for** round $t = 1, \ldots, R$ **do**
2:     Server broadcasts $U^{t-1}$
3:     **for** client $i = 1, \ldots, N$ **in parallel do**
4:         Construct $\tilde{A}_i$ by selecting columns $\mathcal{S}$
5:         $W_{i,0}^t \leftarrow U^{t-1}$
6:         **for** local step $k = 1, \ldots, K$ **do**
7:             $W_{i,k}^t \leftarrow W_{i,k-1}^t - \eta_l \nabla \tilde{\mathcal{L}}_i(W_{i,k-1}^t)$
8:         **end for**
9:         Send $\Delta W_i^t \leftarrow W_{i,K}^t - U^{t-1}$ to server
10:     **end for**
11:     $U^t \leftarrow U^{t-1} + \sum_{i=1}^N \frac{n_i}{n} \Delta W_i^t$
12: **end for**
13: **return** $U^{(R)}$

---

### 4.3 Algorithm Overview

To address these fundamental barriers, we introduce FL-Sailer, a federated learning framework that leverages the inherent low-rank structure of single-cell genomic data to overcome the challenges of ultra-high dimensionality, extreme sparsity, and institutional heterogeneity.

**Phase 1: Federated Dynamic Feature Selection** Each client computes local column leverage scores on ultra high dimensional scATAC-seq data using randomized QR decomposition, providing a communication-efficient alternative to exact SVD while maintaining theoretical guarantees. The server aggregates these scores to construct a global feature sampling distribution that prioritizes biologically informative genomic regions, reducing per-round communication from $O(d)$ to $O(s)$ with $s = \Theta(r \log(r/\delta)/\epsilon^2)$.

**Phase 2: Subspace Federated Training** Clients sample features according to the global distribution and collaboratively train an invariant VAE within the reduced subspace. Our approach combines a chromosome-block architecture with explicit mutual information minimization to disentangle biological signals from technical confounders, enabling robust model aggregation across heterogeneous institutions while preserving privacy.

**Federated Implementation.** In Algorithm 1, clients compute approximate leverage scores locally via randomized projection and transmit only these $O(d)$ scalars. The server aggregates them weighted by sample size to prioritize features that are informative across the federation. This approach exploits the shared genomic coordinate system inherent to scATAC-seq, where feature informativeness reflects conserved regulatory biology rather than institution-specific artifacts.

# 5 Convergence in the Sampled Subspace

**Overview.** This section establishes the end-to-end convergence guarantee for FL-Sailer. We build on the convergence result from Li et al. (2022), which extends convergence to nonconvex objectives under relaxed assumptions. We show that our invariant VAE objective satisfies these assumptions in the sampled subspace. Then we incorporate the subspace embedding guarantees (Section 4) to prove that the sampled problem closely approximates the original problem, yielding explicit convergence rates with bounded optimization and approximation errors.

## 5.1 Key Assumptions and Preliminaries

Our convergence analysis builds on the foundational work of Li et al. (2022), who extended FedAvg convergence theory to non-convex settings via relaxed smoothness conditions. Rather than requiring global smoothness, they established that FedAvg converges at a linear rate up to a variance-dependent error floor under three relaxed local regularity conditions: $(a, b)$-semi-smoothness allows gradient errors scaled by function value, $(\alpha, \beta)$-semi-Lipschitz bounds gradient differences, and $(\tau_1, \tau_2)$-non-critical-point ensures progress near optima. The formal assumptions and the full FedAvg convergence result are provided in Appendix D.1.

To apply this theory to FL-Sailer, we establish the smoothness-related conditions for the sampled objective and adopt the standard local non-critical-point condition used in the underlying FedAvg theory. In Appendix D.2, we prove that the VAE loss inherits the necessary semi-smoothness properties from the original loss, enabling the use of FedAvg in our setting. Additionally, the leverage score sampling (Section 4.1) ensures that the optimization geometry is preserved in the sampled subspace, as formalized in Appendix D.3. These foundations allow us to establish end-to-end convergence for FL-Sailer in Section 5.2.

## 5.2 Convergence of FL-Sailer

The convergence of FL-Sailer is established through a two-part argument: (1) convergence to the sampled problem's solution via FL, and (2) bounding the approximation error introduced by feature sampling relative to the original problem.

**Theorem 5.1** (FL-Sailer Convergence, Main Result). *Consider FL-Sailer with N clients, leverage score sampling rate $\rho = s/d$, and VAE loss satisfying Theorem D.3. Let $\tilde{U}^t$ denote the global model at round t in the sampled space. Under the conditions of Theorem D.1, after R communication rounds:*

$$\mathbb{E}[\tilde{\mathcal{L}}(\tilde{U}^R)] - \mathcal{L}(U^*) \leq \underbrace{(1 - \lambda_1)^R \Delta_0 + 2\lambda_2}_{\text{FedAvg optimization error}} + \underbrace{C \cdot \epsilon \cdot \mathcal{L}(U^*)}_{\text{Sampling approximation error}}$$

*where:*

- $\lambda_1 = \frac{K\eta_l\eta_g}{4}(1 - 4bK\eta_l\eta_g - 2a)\tau_1^2$ *(convergence rate)*
- $\lambda_2 = (1 + a + bK\eta_l\eta_g)\frac{K\eta_l\eta_g}{10}\sigma^2$ *(variance floor)*
- $\Delta_0 = \tilde{\mathcal{L}}(\tilde{U}^0) - \tilde{\mathcal{L}}(\tilde{U}^*)$ *(initial gap in sampled space)*

This establishes that FL-Sailer converges to an $\epsilon$-approximate solution of the original high-dimensional problem with explicit control over both sources of error. The first term represents standard FedAvg optimization error that decays exponentially with rounds $R$, while the second term captures the fixed approximation error due to feature sampling. Remarkably, for high-dimensional genomic data where $d \gg r$, even aggressive sampling with $\rho = 0.2$ yields small approximation error since $\epsilon = O(\sqrt{r/(\rho d)})$ vanishes as $d \to \infty$. In practice, sampling actually improves performance by acting as an implicit regularizer that aligns gradients across heterogeneous clients.

*Proof Sketch.* **Part A: Convergence on the sampled problem.** Since the sampled VAE loss $\tilde{L}$ inherits the semi-smoothness properties from the original loss (Theorem D.3), we can apply Theorem D.1 to obtain:

$$\mathbb{E}[\tilde{\mathcal{L}}(\tilde{U}^R)] - \tilde{\mathcal{L}}(\tilde{U}^*) \leq (1 - \lambda_1)^R \Delta_0 + 2\lambda_2.$$

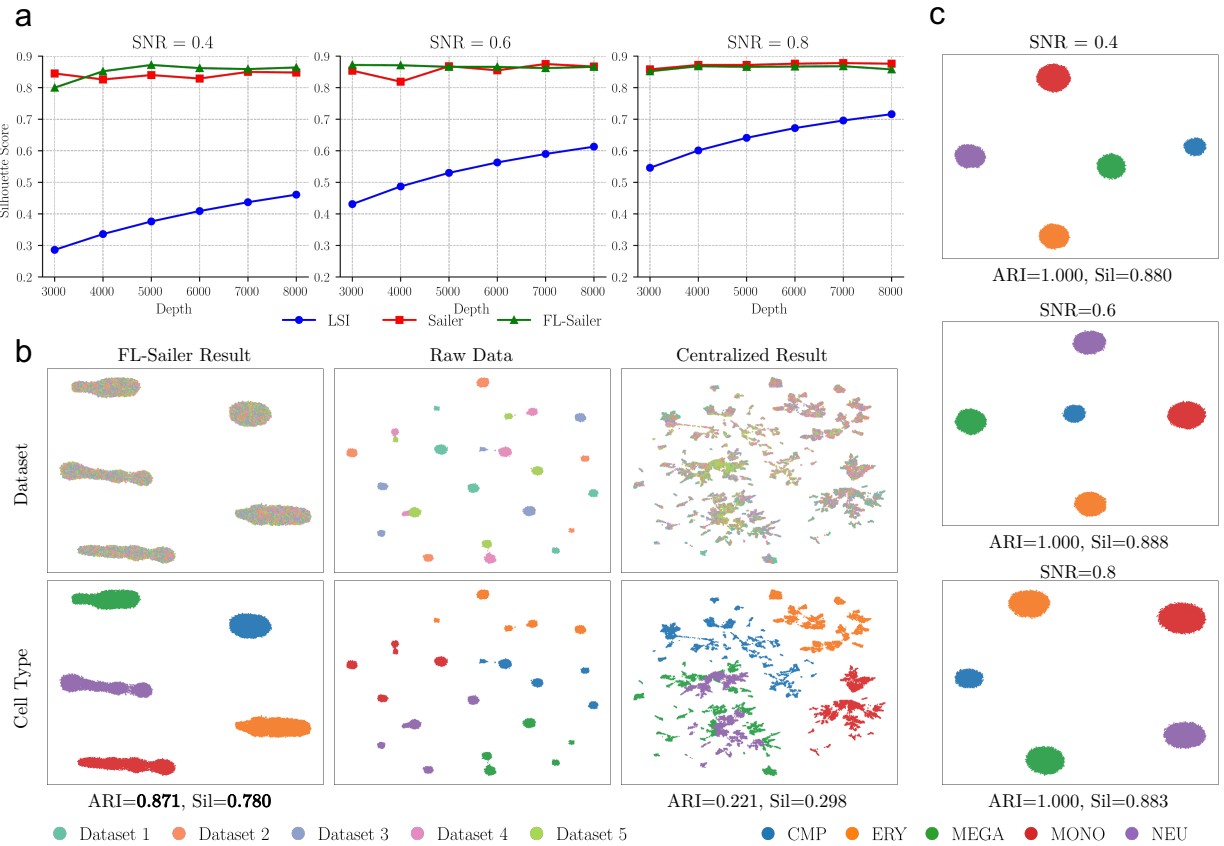

Figure 2: **FL-Sailer overcomes key FL barriers on synthetic scATAC-seq dataset.** (a) Performance under homogeneous conditions: FL-Sailer matches centralized accuracy while preserving privacy. (b) Robustness to confounded heterogeneity: Disentangles biological signals from technical noise. (c) Robustness to extreme class imbalance: Maintains rare cell population detection across SNRs. These results show that FL-Sailer enables federated analysis and turns heterogeneity into a robustness advantage.

**Part B: Approximation quality bound.** Apply the result from Lemma D.5, there is

$$|\tilde{\mathcal{L}}(\tilde{U}^*) - \mathcal{L}(U^*)| \leq C \cdot \epsilon \cdot \mathcal{L}(U^*).$$

Combining Parts A and B via triangle inequality completes the proof. □

We defer the complete proof in Appendix D.4.

## 6 Experiments

**Overview.** This section systematically validates FL-Sailer through three critical stages. We first establish that our federated framework matches centralized performance under homogeneous conditions while demonstrating superior robustness to sequencing depth heterogeneity, confounded variations, and extreme class imbalance (Section 6.1). We then reveal the core breakthrough: on real-world scATAC-seq data, adaptive sampling enables FL to outperform centralized methods while achieving >80% communication reduction (Section 6.2). Finally, comprehensive benchmarking shows FL-Sailer's domain-specific design yields state-of-the-art performance where general-purpose federated algorithms fail (Section 6.3). These experiments establish FL-Sailer as a superior paradigm for collaborative single-cell epigenomics.

## 6.1 FL-Sailer Solves the Problem of Statistical Heterogeneity and Client Drift

We first validate FL-Sailer's fundamental robustness on simulated data (SCAN-ATAC-sim (Chen et al., 2021b); see Appendix E), which allows controlled assessment against key federated challenges.

**Performance Under Homogeneous Conditions** Under ideal, homogeneous conditions (i.e., no batch effects or confounding factors), FL-Sailer achieves clustering performance virtually identical to the centralized Sailer baseline (Figure 2a), with ARI scores of 0.950–1.000 across varying signal-to-noise ratios (SNR) and sequencing depths. This establishes that our federated framework introduces no performance degradation compared to centralized training when data is uniformly distributed, providing a crucial baseline for evaluating its advantages under heterogeneity.

Table 1: **Clustering performance under heterogeneous sequencing depths across clients.** Five clients possess data with varying sequencing depths (3000, 4000, 5000, 6000, and 7000 fragments per cell).

| SNR | Sailer (Centralized) | | FL-Sailer (Federated) | |
|-----|------|------------|------|------------|
| | ARI | Silhouette | ARI | Silhouette |
| 0.4 | $0.620 \pm 0.002$ | $0.214 \pm 0.003$ | $0.858 \pm 0.002$ | $0.740 \pm 0.002$ |
| 0.6 | $0.999 \pm 0.001$ | $0.802 \pm 0.001$ | $0.981 \pm 0.001$ | $0.846 \pm 0.001$ |
| 0.8 | $1.000 \pm 0.000$ | $0.817 \pm 0.001$ | $1.000 \pm 0.000$ | $0.849 \pm 0.001$ |

**Robustness to Heterogeneous Sequencing Depth** Under heterogeneous sequencing depths across clients (3000–7000 fragments/cell), FL-Sailer outperforms centralized training despite the latter's full data access. At low SNR (0.4), FL-Sailer achieves ARI=0.858 versus 0.620 for centralized (38% improvement), with superior silhouette scores (0.740 vs. 0.214; Table 1). While centralized training is confounded by mixing data of varying quality, FL-Sailer leverages this diversity: each client learns robust features locally before aggregation, building a global model resilient to individual client noise.

**Robustness to Confounded Heterogeneity** When technical and biological variations are confounded (each client with unique SNR and cell-type distributions; Table 5 in Appendix F.1), centralized training fails (ARI=0.221) as it learns spurious correlations between noise patterns and cell identity (Figure 2b). In contrast, FL-Sailer achieves successful disentanglement through invariant representation learning ($I(z; c) \leq \epsilon$), yielding clear biological clustering (ARI=0.871)—a 3.9× improvement. This demonstrates that explicit invariance modeling, not mere data availability, enables robust biological discovery under confounded heterogeneity.

**Robustness to Extreme Class Imbalance** FL-Sailer effectively preserves rare cell populations under extreme class imbalance (16-fold difference; CMP: 2.1%, see Appendix F.2). Despite severe non-IID distributions, it achieves perfect clustering (ARI=1.000) across all SNR conditions (Figure 2c). This capability stems from two mechanisms: (1) the invariance objective prevents majority classes from imposing local biases during aggregation, and (2) the chromosome-block architecture maintains gradient flow from minority classes by capturing cell-type-specific regulatory modules, ensuring rare population signals persist through federated training.

## 6.2 FL-Sailer Solves the Communication Bottleneck in High-Dimensional Settings

We evaluate FL-Sailer on three real-world scATAC-seq datasets with extreme dimensionality: Brain PFC (127K features), PsychENCODE (423K features), and PBMC (108K features). Our adaptive sampling strategy transforms the computational barrier of FL into a performance advantage, enabling FL-Sailer to surpass centralized methods with full data access while achieving dramatic communication reduction.

**From Feasibility to Superiority: Synergistic Gains on Real-World Data** FL-Sailer achieves state-of-the-art performance on real-world scATAC-seq data, outperforming both centralized and federated baselines (Figure 3). With adaptive sampling ($\rho = 0.2$), it surpasses the centralized SAILER (Cao et al., 2021)

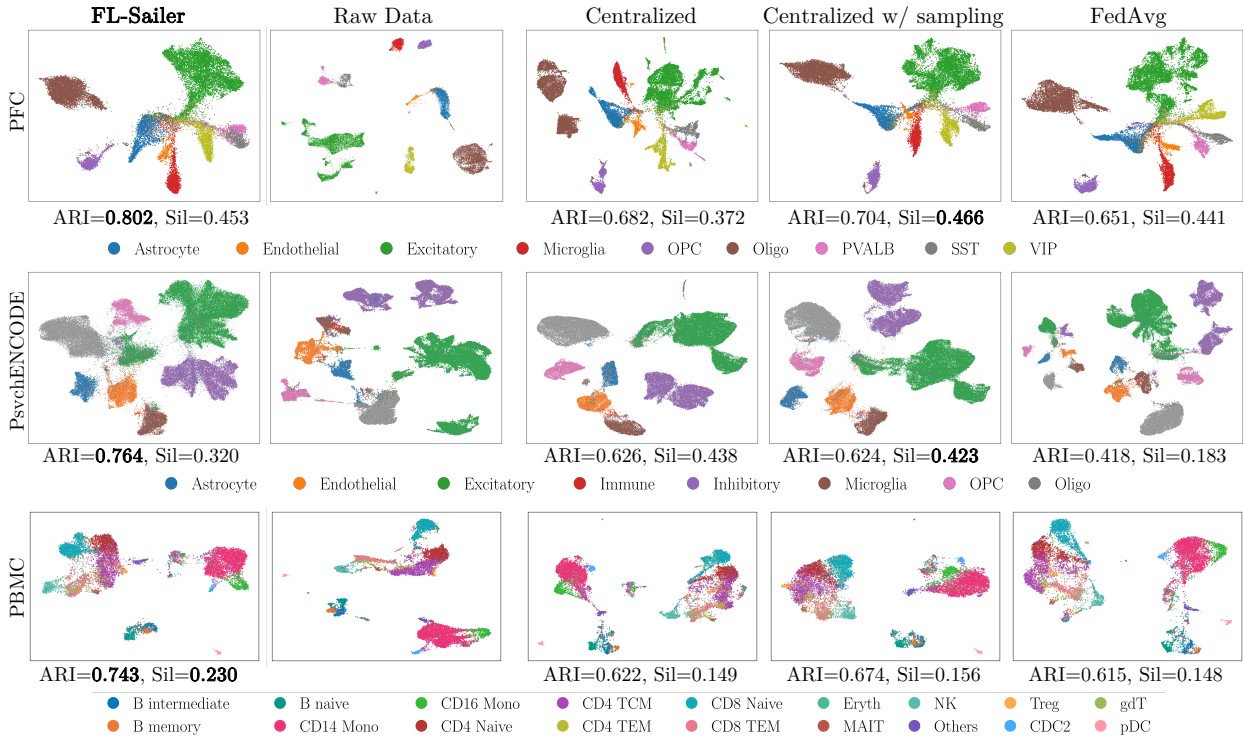

Figure 3: **Comparative analysis of FL-Sailer clustering performance on real world scATAC-seq dataset.** We evaluate four approaches on Brain PFC (127,219 features), PsychENCODE (423,443 features), and PBMC (108,344 features) datasets, visualized using UMAP projections. We compare our proposed FL-Sailer architecture with the raw data and various methods: (1) centralized training, i.e. Sailer's method (Cao et al., 2021); centralized training with adaptive leverage score sampling; FedAvg method (McMahan et al., 2017).

oracle (Brain PFC: 0.802 vs 0.682; PsychENCODE: 0.764 vs 0.626; PBMC: 0.743 vs 0.622) while dramatically outperforming vanilla FedAvg (McMahan et al., 2017) which suffers from performance collapse (e.g., PsychENCODE: 0.418). The key insight is the synergistic effect between FL and adaptive sampling: while offering marginal gains in centralized settings, sampling transforms federated performance by acting as an implicit regularizer that aligns client gradients and suppresses client-specific noise, enabling biological clustering sharper than even the full-data oracle.

**Communication Efficiency Analysis** FL-Sailer achieves approximately 80% communication reduction at the optimal sampling rate ($\rho = 0.2$), transforming prohibitive costs into practical requirements: from 20.40 GB to 4.12 GB for PsychENCODE, 6.16 GB to 1.27 GB for Brain PFC, and 5.26 GB to 1.09 GB for PBMC over 100 rounds (Table 2). Crucially, this biologically-informed sampling simultaneously enhances clustering performance—unlike generic compression methods that typically sacrifice accuracy for efficiency.

**Impact of Sampling Rate** The sampling rate $\rho$ balances communication efficiency and model performance. Table 3 reveals that any sampling ($\rho < 1.0$) dramatically improves federated performance over the no-sampling baseline (e.g., PsychENCODE: $0.418 \rightarrow >0.735$), confirming that feature selection enhances rather than compromises performance. We identify an optimal range of $\rho = 0.15$–$0.30$ across datasets, with peak performance at $\rho = 0.15$ (Brain PFC, ARI=0.855), $\rho = 0.20$ (PsychENCODE, ARI=0.781), and $\rho = 0.30$ (PBMC, ARI=0.743). This demonstrates that retaining only 15–30% of features captures the core biological signal while enabling efficient FL.

## 6.3 Comparison with Standard FL Methods

FL-Sailer consistently and substantially outperforms all FL baselines across three real-world datasets (Table 4). On the most complex PsychENCODE dataset, FL-Sailer achieves an ARI of 0.781, significantly

Table 2: **Complete communication efficiency analysis of FL-Sailer across all sampling rates and datasets.** The table reports model parameters, per-round communication cost, total communication for 100 rounds, and percentage reduction compared to the no-sampling ($\rho = 1.0$) baseline.

| $\rho$ | Brain PFC (127,219 features) | | | | PsychENCODE (423,443 features) | | | | PBMC (108,344 features) | | | |
|---|---|---|---|---|---|---|---|---|---|---|---|---|
| | Params (M) | MB/ round | Total GB | Reduc- tion | Params (M) | MB/ round | Total GB | Reduc- tion | Params (M) | MB/ MB/ | Total Total | Reduc- tion |
| 1.00 | 16.55 | 63.12 | 6.16 | - | 54.75 | 208.87 | 20.40 | - | 14.11 | 53.83 | 5.26 | - |
| 0.50 | 8.34 | 31.82 | 3.11 | 49.6% | 27.44 | 104.68 | 10.22 | 49.9% | 7.12 | 27.17 | 2.65 | 49.5% |
| 0.45 | 7.52 | 28.69 | 2.80 | 54.6% | 24.71 | 94.26 | 9.21 | 54.9% | 6.42 | 24.51 | 2.39 | 54.5% |
| 0.40 | 6.70 | 25.56 | 2.50 | 59.5% | 21.98 | 83.84 | 8.19 | 59.9% | 5.73 | 21.84 | 2.13 | 59.4% |
| 0.35 | 5.88 | 22.43 | 2.19 | 64.5% | 19.25 | 73.42 | 7.17 | 64.8% | 5.03 | 19.17 | 1.87 | 64.4% |
| 0.30 | 5.06 | 19.30 | 1.88 | 69.4% | 16.52 | 63.00 | 6.15 | 69.8% | 4.33 | 16.51 | 1.61 | 69.3% |
| 0.25 | 4.24 | 16.17 | 1.58 | 74.4% | 13.79 | 52.59 | 5.14 | 74.8% | 3.63 | 13.84 | 1.35 | 74.3% |
| **0.20** | **3.42** | **13.03** | **1.27** | **79.4%** | **11.06** | **42.17** | **4.12** | **79.8%** | **2.93** | **11.18** | **1.09** | **79.2%** |
| 0.15 | 2.60 | 9.90 | 0.97 | 84.3% | 8.32 | 31.75 | 3.10 | 84.8% | 2.23 | 8.51 | 0.83 | 84.2% |
| 0.10 | 1.78 | 6.77 | 0.66 | 89.3% | 5.59 | 21.33 | 2.08 | 89.8% | 1.53 | 5.85 | 0.57 | 89.1% |

Table 3: **Impact of the leverage score sampling rate $\rho$ on clustering performance.**

| Dataset | Metric | 0.10 | 0.15 | 0.20 | 0.25 | 0.30 | 0.35 | 0.40 | 0.45 | 0.50 | 1.00 |
|---|---|---|---|---|---|---|---|---|---|---|---|
| Brain PFC | ARI | $0.830 \pm 0.004$ | $\mathbf{0.855 \pm 0.005}$ | $0.802 \pm 0.004$ | $0.781 \pm 0.005$ | $0.769 \pm 0.004$ | $0.772 \pm 0.003$ | $0.732 \pm 0.005$ | $0.730 \pm 0.005$ | $0.734 \pm 0.005$ | $0.651 \pm 0.005$ |
| | Silhouette | $0.424 \pm 0.002$ | $\mathbf{0.463 \pm 0.002}$ | $0.453 \pm 0.002$ | $0.435 \pm 0.002$ | $0.438 \pm 0.002$ | $0.445 \pm 0.002$ | $0.383 \pm 0.002$ | $0.411 \pm 0.003$ | $0.368 \pm 0.002$ | $0.441 \pm 0.003$ |
| PsychENCODE | ARI | $0.734 \pm 0.002$ | $0.740 \pm 0.002$ | $\mathbf{0.781 \pm 0.001}$ | $0.755 \pm 0.002$ | $0.682 \pm 0.002$ | $0.700 \pm 0.002$ | $0.732 \pm 0.002$ | $0.736 \pm 0.002$ | $0.602 \pm 0.002$ | $0.418 \pm 0.002$ |
| | Silhouette | $0.283 \pm 0.001$ | $0.330 \pm 0.001$ | $\mathbf{0.378 \pm 0.001}$ | $0.338 \pm 0.001$ | $0.343 \pm 0.001$ | $0.350 \pm 0.001$ | $0.365 \pm 0.001$ | $0.374 \pm 0.001$ | $0.339 \pm 0.001$ | $0.183 \pm 0.002$ |
| PBMC | ARI | $0.727 \pm 0.006$ | $0.730 \pm 0.006$ | $0.730 \pm 0.006$ | $0.736 \pm 0.006$ | $\mathbf{0.743 \pm 0.005}$ | $0.734 \pm 0.006$ | $0.690 \pm 0.006$ | $0.720 \pm 0.007$ | $0.721 \pm 0.006$ | $0.615 \pm 0.006$ |
| | Silhouette | $0.191 \pm 0.004$ | $0.216 \pm 0.003$ | $0.222 \pm 0.003$ | $0.230 \pm 0.003$ | $\mathbf{0.230 \pm 0.003}$ | $0.224 \pm 0.003$ | $0.196 \pm 0.004$ | $0.226 \pm 0.004$ | $0.218 \pm 0.004$ | $0.148 \pm 0.004$ |

surpassing the next-best method, FedProx (Li et al., 2020) (0.733), and other approaches: FedAvg (McMahan et al., 2017) (0.418), FedOpt (Reddi et al., 2020) (0.482), SCAFFOLD (Karimireddy et al., 2020) (0.655), FedNova (Wang et al., 2020) (0.461), and FedBN (Li et al., 2021) (0.623). This underscores the critical importance of domain-specific design. While general-purpose FL methods (e.g., FedProx, SCAFFOLD) address non-IID data generically, FL-Sailer's synergistic combination of a chromosome-block architecture and invariant representation learning directly captures biological signals while suppressing technical noise, establishing a new state-of-the-art for federated single-cell analysis.

Table 4: **Comparison of federated methods (ARI and Silhouette Scores).**

| Dataset | Metric | FedAvg | FedProx | FedBN | FedOpt | SCAFFOLD | FedNova | FL-Sailer |
|---|---|---|---|---|---|---|---|---|
| Brain PFC | ARI | $0.651 \pm 0.005$ | $0.678 \pm 0.006$ | $0.505 \pm 0.005$ | $0.630 \pm 0.005$ | $0.671 \pm 0.005$ | $0.659 \pm 0.007$ | $\mathbf{0.855 \pm 0.005}$ |
| | Silhouette | $0.441 \pm 0.003$ | $0.307 \pm 0.003$ | $0.147 \pm 0.002$ | $0.426 \pm 0.002$ | $0.386 \pm 0.003$ | $0.432 \pm 0.004$ | $\mathbf{0.463 \pm 0.002}$ |
| PsychENCODE | ARI | $0.418 \pm 0.002$ | $0.731 \pm 0.002$ | $0.623 \pm 0.001$ | $0.482 \pm 0.001$ | $0.654 \pm 0.002$ | $0.461 \pm 0.002$ | $\mathbf{0.781 \pm 0.001}$ |
| | Silhouette | $0.183 \pm 0.002$ | $0.325 \pm 0.001$ | $0.411 \pm 0.001$ | $0.190 \pm 0.001$ | $0.280 \pm 0.001$ | $0.193 \pm 0.002$ | $\mathbf{0.378 \pm 0.001}$ |
| PBMC | ARI | $0.615 \pm 0.006$ | $0.733 \pm 0.006$ | $0.729 \pm 0.005$ | $0.652 \pm 0.007$ | $0.721 \pm 0.007$ | $0.629 \pm 0.008$ | $\mathbf{0.743 \pm 0.005}$ |
| | Silhouette | $0.148 \pm 0.004$ | $0.170 \pm 0.004$ | $0.205 \pm 0.003$ | $0.152 \pm 0.003$ | $0.160 \pm 0.003$ | $0.148 \pm 0.005$ | $\mathbf{0.230 \pm 0.003}$ |

# 7 Conclusion

We present FL-Sailer, a federated learning framework that overcomes scalability and heterogeneity barriers in scATAC-seq analysis via adaptive sampling and invariant representation learning with theoretical guarantee. It enables efficient, privacy-preserving multi-institutional studies and even outperforms centralized methods, establishing a new paradigm for collaborative genomics.

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

## A  Algorithmic Implementation of Randomized Leverage Score Sampling

---
**Algorithm 2** Randomized Column Leverage Score Sampling

---
**Require:** Matrix $A \in \mathbb{R}^{n \times d}$, sketch size $s_k$, number of columns to sample $s$.
1: Generate random matrix $\Omega \leftarrow \text{randn}(s_k, n)$
2: $B \leftarrow \Omega A \in \mathbb{R}^{s_k \times d}$
3: Compute QR factorization: $B^{\top} = QR$
4: **for** $j = 1, \ldots, d$ **do**
5: $\quad \hat{\ell}_j \leftarrow \|Q_{j,:}\|_2^2$
6: **end for**
7: $L_{\text{sum}} \leftarrow \sum_{j=1}^{d} \hat{\ell}_j$
8: **for** $j = 1, \ldots, d$ **do**
9: $\quad p_j \leftarrow \hat{\ell}_j / L_{\text{sum}}$
10: **end for**
11: Sample $S = \{j_1, \ldots, j_s\} \subset \{1, \ldots, d\}$ of size $s$ using weighted sampling without replacement based on probabilities $p$
12: Construct $\tilde{A}$ by selecting columns indexed by $S$
13: **return** $\tilde{A}$

---

In this appendix, we detail the randomized sketching technique used in Phase 1 of Algorithm 1. Computing exact leverage scores requires singular value decomposition (SVD), which incurs a prohibitive computational cost of $O(n^2 d)$ for high-dimensional scATAC-seq data.

To overcome this, Algorithm 2 employs a randomized projection method (Drineas et al., 2012). The key idea is to project the input matrix $A \in \mathbb{R}^{n \times d}$ onto a lower-dimensional subspace using a Gaussian random matrix $\Omega \in \mathbb{R}^{s_k \times n}$, where $s_k \ll n$ is the sketch size. This projection preserves the geometry of the column space, allowing us to approximate the leverage scores using the orthogonal basis of the sketched matrix.

**Complexity Analysis.** The dominant operations are the matrix multiplication $B = \Omega A$ and the QR decomposition of the reduced matrix. This reduces the overall time complexity to $O(\text{nnz}(A)s_k + ds_k^2)$, which is linear in both the number of cells $n$ and features $d$, making it highly efficient for client-side execution.

## B  Theoretical Guarantees of Adaptive Feature Sampling

### B.1  Preservation of Optimization Geometry in the Reduced Subspace

**Remark B.1** (Optimization Objective under Feature Sampling). *Consider the original optimization problem:* $\min_{w \in \mathbb{R}^d} \|Aw - y\|_2^2$, *After feature sampling with selection matrix $T \in \mathbb{R}^{d \times s}$ (where $s \ll d$), we obtain the reduced matrix $\tilde{A} = AT \in \mathbb{R}^{n \times s}$. The optimization problem becomes:*

$$\min_{\tilde{w} \in \mathbb{R}^s} \|\tilde{A}\tilde{w} - y\|_2^2 = \min_{\tilde{w} \in \mathbb{R}^s} \|AT\tilde{w} - y\|_2^2,$$

*where $\tilde{w} \in \mathbb{R}^s$ is the parameter vector in the reduced feature space.*

*From Lemma 4.1, the relationship between the original and reduced problems is characterized as follows:*

1. *Let $w^* = \arg\min_w \|Aw - y\|_2^2 = A^\dagger y$ be the optimal solution in the original space, and $\tilde{w}^* = \arg\min_{\tilde{w}} \|AT\tilde{w} - y\|_2^2 = (AT)^\dagger y$ be the optimal solution in the reduced space.*

2. *The reconstruction $w_{recon} = T\tilde{w}^*$ provides an exact solution in terms of objective value. Under the assumptions of Lemma 4.1, we have*

$$\|Aw_{recon} - y\|_2^2 = \|Aw^* - y\|_2^2.$$

3. *The gradient information is preserved: for any $\tilde{w} \in \mathbb{R}^s$, if $\nabla f(w) = 2A^\top(Aw - y)$ and $\nabla \tilde{f}(\tilde{w}) = 2(AT)^\top(AT\tilde{w} - y)$, there is*

$$(1 - \epsilon)\|\nabla f(T\tilde{w})\|_2^2 \leq \|\nabla \tilde{f}(\tilde{w})\|_2^2 \leq (1 + \epsilon)\|\nabla f(T\tilde{w})\|_2^2$$

**Remark B.2** (Federated Leverage Score Aggregation). *The weighted aggregation in Algorithm 1 computes an approximation to the global leverage scores. For scATAC-seq data, all institutions profile identical genomic regions governed by conserved regulatory programs, which induces shared column space structure across clients. This biological constraint ensures that the aggregated scores preserve the relative ranking of feature importance, a property sufficient for selecting informative regulatory regions. Section 6 empirically validates this approach across diverse real-world datasets.*

### B.1.1 Proof of Remark B.1, Part 2

*Proof.* Let $\tilde{A} = AT$. We define the optimal solutions for the original and sampled problems respectively as:

$$w^* := \arg\min_{w \in \mathbb{R}^d} \|Aw - y\|_2^2, \qquad \tilde{w}^* := \arg\min_{\tilde{w} \in \mathbb{R}^s} \|\tilde{A}\tilde{w} - y\|_2^2.$$

Define the reconstructed vector $w_{\text{recon}} := T\tilde{w}^*$, so that $Aw_{\text{recon}} = AT\tilde{w}^* = \tilde{A}\tilde{w}^*$.

First, we establish a lower bound. We have:

$$\|Aw^* - y\|_2^2 \leq \|Aw_{\text{recon}} - y\|_2^2 = \|\tilde{A}\tilde{w}^* - y\|_2^2, \tag{3}$$

where the inequality follows from the fact that $w^*$ is the global minimizer of the original objective $\|Aw - y\|_2^2$, and the equality follows from the definition $Aw_{\text{recon}} = \tilde{A}\tilde{w}^*$.

Next, we establish the upper bound. By Lemma 4.1, with probability $1 - \delta$, for all $v \in \mathbb{R}^n$:

$$(1 - \epsilon)\|A^\top v\|_2^2 \leq \|\tilde{A}^\top v\|_2^2 \leq (1 + \epsilon)\|A^\top v\|_2^2. \tag{4}$$

This subspace embedding property implies that the column spaces of the original and sampled matrices are identical:

$$\text{col}(\tilde{A}) = \ker(\tilde{A}^\top)^\perp = \ker(A^\top)^\perp = \text{col}(A),$$

where the first and last equalities follow from the fundamental theorem of linear algebra, and the middle equality follows from (4) which implies $\ker(A^\top) = \ker(\tilde{A}^\top)$ (since for $\epsilon \in (0, 1)$, $\|A^\top v\|_2 = 0$ if and only if $\|\tilde{A}^\top v\|_2 = 0$).

Since $Aw^* \in \text{col}(A)$, the column space equality implies $Aw^* \in \text{col}(\tilde{A})$. Therefore, there exists a vector $u \in \mathbb{R}^s$ (e.g., $u = \tilde{A}^\dagger Aw^*$) such that:

$$\tilde{A}u = Aw^*. \tag{5}$$

Using this vector $u$, we can bound the objective value of the sampled problem:

$$\begin{aligned}\|\tilde{A}\tilde{w}^* - y\|_2 &\leq \|\tilde{A}u - y\|_2 \\ &= \|Aw^* - y\|_2,\end{aligned} \tag{6}$$

where the inequality follows from the optimality of $\tilde{w}^*$ for the problem $\min_{\tilde{w}} \|\tilde{A}\tilde{w} - y\|_2^2$, and the equality follows from substituting (5).

Finally, combining the bounds yields the result:

$$\|Aw_{\mathrm{recon}} - y\|_2^2 = \|Aw^* - y\|_2^2,$$

where the equality follows from combining the lower bound in (3) and the upper bound in (6). This demonstrates that under the subspace embedding assumption, the reconstruction error matches the original optimal error exactly.

$\square$

### B.1.2 Proof of Remark B.1, Part 3

*Proof.* For any $\tilde{w} \in \mathbb{R}^s$, let $f(w) = \|Aw - y\|_2^2$ and $\tilde{f}(\tilde{w}) = \|AT\tilde{w} - y\|_2^2$. We aim to prove

$$(1 - \epsilon)\|\nabla f(T\tilde{w})\|_2^2 \le \|\nabla \tilde{f}(\tilde{w})\|_2^2 \le (1 + \epsilon)\|\nabla f(T\tilde{w})\|_2^2$$

Here the gradients are

$$\nabla f(w) = 2A^\top(Aw - y)$$
$$\nabla \tilde{f}(\tilde{w}) = 2(AT)^\top(AT\tilde{w} - y).$$

For the composed function at $w = T\tilde{w}$, we have $\nabla f(T\tilde{w}) = 2A^\top(AT\tilde{w} - y)$. Therefore,

$$\|\nabla \tilde{f}(\tilde{w})\|_2^2 = 4\|(AT)^\top(AT\tilde{w} - y)\|_2^2;$$
$$\|\nabla f(T\tilde{w})\|_2^2 = 4\|A^\top(AT\tilde{w} - y)\|_2^2.$$

Let $e := AT\tilde{w} - y \in \mathbb{R}^n$. By Lemma 4.1, for all $v \in \mathbb{R}^n$,

$$(1 - \epsilon)\|A^\top v\|_2^2 \le \|(AT)^\top v\|_2^2 \le (1 + \epsilon)\|A^\top v\|_2^2.$$

Applying this with $v = e$ yields

$$(1 - \epsilon)\|A^\top(AT\tilde{w} - y)\|_2^2 \le \|(AT)^\top(AT\tilde{w} - y)\|_2^2$$
$$\le (1 + \epsilon)\|A^\top(AT\tilde{w} - y)\|_2^2.$$

Multiplying by 4 gives

$$(1 - \epsilon)\|\nabla f(T\tilde{w})\|_2^2 \le \|\nabla \tilde{f}(\tilde{w})\|_2^2 \le (1 + \epsilon)\|\nabla f(T\tilde{w})\|_2^2.$$

This completes the proof. $\square$

## B.2 Preservation of Biological Structure and Marker Gene Informativeness

This section proves that our adaptive sampling preserves biological signal. While the previous section showed it maintains optimization geometry, Theorem B.3 guarantees it also retains the data's biological structure. Specifically, we prove the sampling (i) preserves distances between cell types, (ii) maintains clustering structure, and (iii) preferentially selects discriminative genomic regions. This ensures that dimensionality reduction removes noise while keeping features necessary for interpretable biological discovery.

**Theorem B.3** (Biological Signal Preservation under Leverage Score Sampling). *Let $A \in \mathbb{R}^{n \times d}$ be a scATAC-seq data matrix where rows represent cells and columns represent genomic features. Assume cells belong to $\mathcal{K}$ distinct cell types $\mathcal{C} = \{C_1, C_2, \ldots, C_\mathcal{K}\}$.*

*For a leverage score sampling transformation $T : \mathbb{R}^d \to \mathbb{R}^s$ (where $s = \rho d$ and $\rho$ is the sampling rate) satisfying the conditions of Remark B.1, the following properties hold:*

1. **Cell Type Separability Preservation**: *For any two distinct cell types $C_i, C_j$, define their separation in the original space as $\Delta_{ij} = \|\mu_i - \mu_j\|_2 / \sqrt{\sigma_i^2 + \sigma_j^2}$ where $\mu_i, \mu_j$ are class centers and $\sigma_i^2, \sigma_j^2$ are within-class variances. The separation after sampling $\tilde{\Delta}_{ij}$ satisfies:*

$$\mathbb{P}\left[(1 - 2\epsilon)\Delta_{ij} \leq \tilde{\Delta}_{ij} \leq (1 + 2\epsilon)\Delta_{ij}\right] \geq 1 - \delta.$$

2. **Clustering Structure Preservation**: *Define the Davies-Bouldin index as $DB = \frac{1}{\mathcal{K}} \sum_{i=1}^{\mathcal{K}} \max_{j \neq i} \frac{\sigma_i + \sigma_j}{\|\mu_i - \mu_j\|_2}$. Then the DB index before and after sampling satisfies:*

$$|DB_{sampled} - DB_{original}| \leq 2\epsilon \cdot DB_{original}.$$

3. **Biological Marker Preservation**: *If genomic region $j$ is a marker for cell type $C_i$ (i.e., its average accessibility in $C_i$ is significantly higher than in other types), then its selection probability satisfies:*

$$\pi_j \geq \min\left\{1, \frac{s}{r} \cdot \frac{\|a_{C_i,j} - a_{\neg C_i,j}\|_2^2}{\sum_{\ell=1}^{d} \|a_{C_i,\ell} - a_{\neg C_i,\ell}\|_2^2}\right\}$$

*where $a_{C_i,j}$ denotes the average accessibility of cell type $C_i$ at region $j$.*

### B.2.1 Proof of Theorem B.3, Part 1

*Proof.* We aim to prove that the cell type separability measure is preserved under leverage score sampling with high probability.

Let $A \in \mathbb{R}^{n \times d}$ be the data matrix where rows represent cells and columns represent genomic features. For each cell type $C_i \subseteq [n]$, define the class center $\mu_i = \frac{1}{|C_i|} \sum_{i' \in C_i} a_{i'}^\top \in \mathbb{R}^d$ where $a_{i'}^\top$ is the row of $A$ corresponding to cell $i'$ the within-class variance $\sigma_i^2 = \frac{1}{|C_i|} \sum_{k \in C_i} \|a_k - \mu_i\|_2^2$, and the original separability $\Delta_{ij} = \frac{\|\mu_i - \mu_j\|_2}{\sqrt{\sigma_i^2 + \sigma_j^2}}$.

After leverage score sampling with selection matrix $T \in \mathbb{R}^{d \times s}$, we obtain the sampled data $\tilde{A} = AT \in \mathbb{R}^{n \times s}$, sampled class center $\tilde{\mu}_i = \frac{1}{|C_i|} \sum_{k \in C_i} a_k^\top T = \mu_i^\top T \in \mathbb{R}^s$, sampled variance $\tilde{\sigma}_i^2 = \frac{1}{|C_i|} \sum_{k \in C_i} \|a_k^\top T - \mu_i^\top T\|_2^2$, and sampled separability $\tilde{\Delta}_{ij} = \frac{\|\tilde{\mu}_i - \tilde{\mu}_j\|_2}{\sqrt{\tilde{\sigma}_i^2 + \tilde{\sigma}_j^2}}$.

First, we apply the subspace embedding property to the class center difference. For any vector $v$ in the row space of $A$, we have with probability at least $1 - \delta$:

$$(1 - \epsilon)\|v\|_2^2 \leq \|vT\|_2^2 \leq (1 + \epsilon)\|v\|_2^2. \tag{7}$$

Since $\mu_i - \mu_j = \frac{1}{|C_i|} \sum_{k \in C_i} a_k^\top - \frac{1}{|C_j|} \sum_{\ell \in C_j} a_\ell^\top$ is a linear combination of rows of $A$, it lies in the row space of $A$. Therefore:

$$(1 - \epsilon)\|\mu_i - \mu_j\|_2^2 \leq \|(\mu_i - \mu_j)T\|_2^2 = \|\tilde{\mu}_i - \tilde{\mu}_j\|_2^2 \leq (1 + \epsilon)\|\mu_i - \mu_j\|_2^2, \tag{8}$$

where the inequalities follow from applying (7) with $v = \mu_i - \mu_j$, and the equality follows from the linearity of matrix multiplication. Taking square roots of (8) yields:

$$\sqrt{1 - \epsilon}\|\mu_i - \mu_j\|_2 \leq \|\tilde{\mu}_i - \tilde{\mu}_j\|_2 \leq \sqrt{1 + \epsilon}\|\mu_i - \mu_j\|_2. \tag{9}$$

Next, we analyze the within-class variances. For each $i' \in C_i$, the vector $(a_{i'} - \mu_i)^\top$ is in the row space of $A$ since $(a_{i'} - \mu_i)^\top = a_{i'}^\top - \frac{1}{|C_i|} \sum_{\ell \in C_i} a_\ell^\top$. In the event of (7), there is

$$(1 - \epsilon)\|a_{i'} - \mu_i\|_2^2 \leq \|(a_{i'} - \mu_i)T\|_2^2 \leq (1 + \epsilon)\|a_{i'} - \mu_i\|_2^2. \tag{10}$$

Since this two-sided bounds holds simultaneously for all $i' \in C_i$,

$$\tilde{\sigma}_i^2 = \frac{1}{|C_i|} \sum_{i' \in C_i} \|(a_{i'} - \mu_i)T\|_2^2$$

$$\in \left[ \frac{1}{|C_i|} \sum_{i' \in C_i} (1 - \epsilon) \|a_{i'} - \mu_i\|_2^2, \frac{1}{|C_i|} \sum_{i' \in C_i} (1 + \epsilon) \|a_{i'} - \mu_i\|_2^2 \right]$$

$$= [(1 - \epsilon)\sigma_i^2, (1 + \epsilon)\sigma_i^2], \tag{11}$$

where the second step follows from applying the bounds in (10) pointwise, and the third step follows from the definition of $\sigma_i^2$. Similarly, under the event of (7), we have $(1 - \epsilon)\sigma_j^2 \le \tilde{\sigma}_j^2 \le (1 + \epsilon)\sigma_j^2$.

Therefore, as (7) holds with probability at least $1 - \delta$, we can give a two-sided bound to the sampled separability $\tilde{\Delta}_{ij} = \frac{\|\tilde{\mu}_i - \tilde{\mu}_j\|_2}{\sqrt{\tilde{\sigma}_i^2 + \tilde{\sigma}_j^2}}$. For the lower bound:

$$\tilde{\Delta}_{ij} \ge \frac{\sqrt{1 - \epsilon} \cdot \|\mu_i - \mu_j\|_2}{\sqrt{(1 + \epsilon)(\sigma_i^2 + \sigma_j^2)}}$$

$$= \sqrt{\frac{1 - \epsilon}{1 + \epsilon}} \cdot \frac{\|\mu_i - \mu_j\|_2}{\sqrt{\sigma_i^2 + \sigma_j^2}}$$

$$= \sqrt{\frac{1 - \epsilon}{1 + \epsilon}} \cdot \Delta_{ij}, \tag{12}$$

where the first step follows from the lower bound in (9) and the upper bound in (11), and the second step follows from algebraic manipulation. Similar result also holds for the upper bound. As a result, there is

$$\sqrt{\frac{1 - \epsilon}{1 + \epsilon}} \cdot \Delta_{ij} \le \tilde{\Delta}_{ij} \le \sqrt{\frac{1 + \epsilon}{1 - \epsilon}} \cdot \Delta_{ij}. \tag{13}$$

Finally, we refine the bounds in (12) and (13). For $\epsilon \in (0, 1)$, we have:

$$\sqrt{\frac{1 - \epsilon}{1 + \epsilon}} = \sqrt{1 - \epsilon} \cdot (1 + \epsilon)^{-1/2}$$

$$\ge (1 - \epsilon/2)(1 - \epsilon/2)$$

$$= 1 - \epsilon + \epsilon^2/4$$

$$\ge 1 - \epsilon,$$

where the second step follows from the inequalities $\sqrt{1 - x} \ge 1 - x/2$ and $(1 + x)^{-1/2} \ge 1 - x/2$ for $x \in [0, 1]$, and the third step follows from expanding the product. Similarly, for $\epsilon \in (0, 1/2)$, $\sqrt{\frac{1 + \epsilon}{1 - \epsilon}} \le 1 + 2\epsilon$. Hence we have

$$\mathbb{P}\left[ (1 - 2\epsilon)\Delta_{ij} \le \tilde{\Delta}_{ij} \le (1 + 2\epsilon)\Delta_{ij} \right] \ge 1 - \delta.$$

This completes the proof. $\square$

### B.2.2 Proof of Theorem B.3, Part 2

*Proof.* We prove that $|DB_{\text{sampled}} - DB_{\text{original}}| \le C \cdot \epsilon \cdot DB_{\text{original}}$ for some constant $C$, where the Davies-Bouldin index is defined as $DB = \frac{1}{\mathcal{K}} \sum_{i=1}^{\mathcal{K}} \max_{j \ne i} R_{ij}$, with $R_{ij} = \frac{\sigma_i + \sigma_j}{\|\mu_i - \mu_j\|_2}$.

Our strategy is to bound the ratio $\tilde{R}_{ij}/R_{ij}$. From the proof of Part 1, we have already established the following bounds which hold simultaneously with high probability:

1. Bound on inter-class distance (from (9)):

$$\sqrt{1 - \epsilon}\|\mu_i - \mu_j\|_2 \le \|\tilde{\mu}_i - \tilde{\mu}_j\|_2 \le \sqrt{1 + \epsilon}\|\mu_i - \mu_j\|_2.$$

2. Bound on intra-class standard deviation (from (11) after taking square root):

$$\sqrt{1-\epsilon}\,\sigma_i \leq \tilde{\sigma}_i \leq \sqrt{1+\epsilon}\,\sigma_i.$$

Now, we combine these bounds to analyze $\tilde{R}_{ij} = \frac{\tilde{\sigma}_i + \tilde{\sigma}_j}{\|\tilde{\mu}_i - \tilde{\mu}_j\|_2}$.

For the upper bound of $\tilde{R}_{ij}$:

$$\tilde{R}_{ij} = \frac{\tilde{\sigma}_i + \tilde{\sigma}_j}{\|\tilde{\mu}_i - \tilde{\mu}_j\|_2} \leq \frac{\sqrt{1+\epsilon}(\sigma_i + \sigma_j)}{\sqrt{1-\epsilon}\|\mu_i - \mu_j\|_2} = \sqrt{\frac{1+\epsilon}{1-\epsilon}} \cdot R_{ij}.$$

For the lower bound of $\tilde{R}_{ij}$:

$$\tilde{R}_{ij} \geq \frac{\sqrt{1-\epsilon}(\sigma_i + \sigma_j)}{\sqrt{1+\epsilon}\|\mu_i - \mu_j\|_2} = \sqrt{\frac{1-\epsilon}{1+\epsilon}} \cdot R_{ij}.$$

As established in the proof of Part 1, for $\epsilon \in (0, 1/2)$, we have $\sqrt{\frac{1+\epsilon}{1-\epsilon}} \leq 1 + 2\epsilon$ and $\sqrt{\frac{1-\epsilon}{1+\epsilon}} \geq 1 - \epsilon$. Combining these, we get:

$$(1 - \epsilon)R_{ij} \leq \tilde{R}_{ij} \leq (1 + 2\epsilon)R_{ij}.$$

Since the max operation preserves these relative bounds, for each $i$:

$$(1 - \epsilon)\max_{j \neq i} R_{ij} \leq \max_{j \neq i} \tilde{R}_{ij} \leq (1 + 2\epsilon)\max_{j \neq i} R_{ij}.$$

Summing over all $i$ and dividing by $\mathcal{K}$ gives the bound for the DB index:

$$(1 - \epsilon)DB_{\text{original}} \leq DB_{\text{sampled}} \leq (1 + 2\epsilon)DB_{\text{original}}.$$

This implies that $|DB_{\text{sampled}} - DB_{\text{original}}|$ is bounded by a term proportional to $\epsilon \cdot DB_{\text{original}}$, which completes the proof. $\qquad\square$

### B.2.3 Proof of Theorem B.3, Part 3

*Proof.* We prove that if genomic region $j$ is a marker for cell type $C_i$, then its selection probability (inclusion probability) $\pi_j$ satisfies the lower bound behavior proportional to its discriminative power.

Let $h \in \mathbb{R}^n$ be a vector that captures the cell type distinction:

$$h_i = \begin{cases} \frac{1}{|C_i|} & \text{if cell } i \in C_i \\ -\frac{1}{|\neg C_i|} & \text{if cell } i \in \neg C_i \end{cases} \tag{14}$$

where $w$ has positive weight for cells in type $C_i$ and negative weight for cells not in $C_i$.

The discriminative score for feature $j$ is:

$$d_j = h^\top a_{:,j} = \sum_{i=1}^n h_i a_{ij} = a_{C_i,j} - a_{\neg C_i,j}, \tag{15}$$

where the first equality follows from the definition of matrix-vector multiplication, and the second equality follows from the specific structure of $h$ in (14) and the definition of average accessibility.

Let $A = U\Sigma V^\top$ be the SVD of $A$. Since $a_{:,j}$ lies in the column space of $A$, we can write:

$$d_j = h^\top a_{:,j} = h^\top U U^\top a_{:,j} = \sum_{i=1}^r (h^\top u_i)(u_i^\top a_{:,j})$$

$$= \sum_{i=1}^{r} (h^\top u_i)\sigma_i \cdot \frac{u_i^\top a_{:,j}}{\sigma_i}, \tag{16}$$

where the first equality follows from $UU^\top$ being the projection onto the column space of $A$, the second equality follows from expanding the matrix multiplication, and the third equality follows from multiplying and dividing by $\sigma_i$ for each term.

By the Cauchy-Schwarz inequality applied to (16):

$$d_j^2 \le \left( \sum_{i=1}^{r} (h^\top u_i)^2 \sigma_i^2 \right) \cdot \left( \sum_{i=1}^{r} \frac{(u_i^\top a_{:,j})^2}{\sigma_i^2} \right), \tag{17}$$

where we set $x_i = (h^\top u_i)\sigma_i$ and $y_i = \frac{u_i^\top a_{:,j}}{\sigma_i}$.

The first term in (17) represents the total discriminative signal:

$$
\begin{aligned}
\sum_{i=1}^{r} (h^\top u_i)^2 \sigma_i^2 &= h^\top U \Sigma^2 U^\top h \\
&= h^\top A A^\top h \\
&= \|A^\top h\|_2^2 \\
&= \sum_{\ell=1}^{d} (h^\top a_{:,\ell})^2 \\
&= \sum_{\ell=1}^{d} d_\ell^2 \\
&= \sum_{\ell=1}^{d} \|a_{C_i,\ell} - a_{\neg C_i,\ell}\|_2^2,
\end{aligned}
\tag{18}
$$

where the first equality follows from the SVD $AA^\top = U\Sigma^2 U^\top$, the second equality follows from the identity $\|A^\top h\|_2^2 = h^\top A A^\top h$, and the last equality follows from the definition of $d_\ell$ in (15).

The second term in (17) is exactly the leverage score:

$$\sum_{i=1}^{r} \frac{(u_i^\top a_{:,j})^2}{\sigma_i^2} = a_{:,j}^\top U \Sigma^{-2} U^\top a_{:,j} = a_{:,j}^\top (AA^\top)^\dagger a_{:,j} = \ell_j^{\mathrm{col}}, \tag{19}$$

where the equalities follow from the properties of the pseudoinverse $(AA^\top)^\dagger = U\Sigma^{-2}U^\top$ and the definition of column leverage score in Lemma 4.1.

Combining (17), (18), and (19), we obtain:

$$d_j^2 \le \left( \sum_{\ell=1}^{d} d_\ell^2 \right) \cdot \ell_j^{\mathrm{col}}. \tag{20}$$

Rearranging (20) gives a lower bound on the leverage score:

$$\ell_j^{\mathrm{col}} \ge \frac{d_j^2}{\sum_{\ell=1}^{d} d_\ell^2} = \frac{\|a_{C_i,j} - a_{\neg C_i,j}\|_2^2}{\sum_{\ell=1}^{d} \|a_{C_i,\ell} - a_{\neg C_i,\ell}\|_2^2}. \tag{21}$$

Finally, we relate leverage scores to the inclusion probability. From Lemma 4.1, define the (base) sampling weights $q_j := \ell_j^{\mathrm{col}}/r$. Let $\pi_j := \mathbb{P}(j \in \mathcal{S})$ be the probability that feature $j$ is included in the sampled set $\mathcal{S}$ of size $s$ produced by weighted sampling without replacement.

Under successive weighted sampling without replacement, at each draw the conditional probability of selecting $j$ given that it has not been selected is at least $q_j$. Hence

$$\mathbb{P}(j \notin \mathcal{S}) \leq (1 - q_j)^s,$$

so

$$
\begin{aligned}
\pi_j &\geq 1 - (1 - q_j)^s \geq 1 - e^{-sq_j} \\
&= 1 - \exp\left(-\frac{s}{r}\ell_j^{\mathrm{col}}\right).
\end{aligned}
$$

Combining with (21), we obtain

$$
\begin{aligned}
\pi_j &\geq 1 - \exp\left(-\frac{s}{r} \cdot \frac{d_j^2}{\sum_{\ell=1}^d d_\ell^2}\right) \\
&= 1 - \exp\left(-\frac{s}{r} \cdot \frac{(a_{C_i,j} - a_{\neg C_i,j})^2}{\sum_{\ell=1}^d (a_{C_i,\ell} - a_{\neg C_i,\ell})^2}\right).
\end{aligned}
$$

In particular, using $1 - e^{-x} \geq (1 - e^{-1})\min\{1, x\}$ for $x \geq 0$ yields the proportional lower bound.

$\square$

## C  Derivation and Regularity Analysis of the Invariant VAE Objective

### C.1  Variational Upper Bound Derivation for Disentanglement

This section provides the complete theoretical derivations for the invariant VAE component, which builds upon the SAILER framework (Cao et al., 2021) and adapts it to the federated learning setting.

**Definition C.1** (Mutual Information). *For random variables $X$, $Y$ with joint distribution $p(x, y)$, their mutual information is defined as: $I(X; Y) = \mathbb{E}_{p(x,y)}\left[\log \frac{p(x,y)}{p(x)p(y)}\right]$.*

Our goal is to minimize the mutual information $I(z; c)$ between learned representations $z$ and confounding variables $c$, while maintaining the model's primary learning objectives. Using variational inference techniques and Lemma C.2, we can show following upper bound for $I(z; c)$.

**Lemma C.2** (Mutual Information Decomposition). *For the Markov chain $c \to x \to z$, the mutual information $I(z; c)$ can be decomposed as:*

$$I(z; c) = I(z; x) - I(z; x|c)$$

*Proof.* Starting with the chain rule of mutual information:

$$I(z; c) = I(z; x) - I(z; x|c) + I(z; c|x).$$

By the Markov chain property, we have $I(z; c|x) = 0$ since $z$ depends only on $x$. Therefore:

$$I(z; c) = I(z; x) - I(z; x|c).$$

$\square$

**Theorem C.3** (Variational Upper Bound). *For any variational encoder distribution $q_\theta(z|x)$ and decoder distribution $p_\phi(x|z, c)$, we have:*

$$I(z; c) \leq \mathbb{E}_{x \sim q(x)}\left[\mathrm{KL}[q_\theta(z|x) \| q_\theta(z)]\right] - H(x|c) - \mathbb{E}_{x,c \sim q(x,c), z \sim q_\theta(z|x)}\left[\log p_\phi(x|z, c)\right] \quad (22)$$

*where $q(x), q(x, c)$ are empirical distribution with respect to $x$ and $(x, c)$. Note that $H(x|c)$ would be a constant with respect to our optimization.*

*Proof of Theorem C.3.* Starting with the decomposition from Lemma C.2, we express the mutual information $I(z; c)$ as:

$$
\begin{aligned}
I(z; c) &= I(z; x) - I(z; x|c) \\
&= I(z; x) - H(x|c) + H(x|z, c),
\end{aligned} \tag{23}
$$

where the first step follows from the chain rule of mutual information, and the second step follows from the definition of conditional mutual information $I(z; x|c) = H(x|c) - H(x|z, c)$.

We analyze the terms in Eq. (23) separately. First, the mutual information term $I(z; x)$ is defined as the KL divergence between the joint and the product of marginals, averaged over $x$:

$$
I(z; x) = \mathbb{E}_{x \sim q(x)} \left[ \mathrm{KL}(q_\theta(z|x) \| q_\theta(z)) \right], \tag{24}
$$

where $q_\theta(z) = \mathbb{E}_{x \sim q(x)}[q_\theta(z|x)]$ is the aggregated posterior.

Next, we bound the conditional entropy term $H(x|z, c)$. Let $q(x|z, c)$ denote the true (intractable) inverse posterior induced by the data distribution $q(x, c)$ and the encoder $q_\theta(z|x)$. For any variational decoder $p_\phi(x|z, c)$, the KL divergence is non-negative:

$$
\mathrm{KL}(q(x|z, c) \| p_\phi(x|z, c)) \geq 0.
$$

Expanding the KL divergence definition, we have:

$$
\mathbb{E}_{x \sim q(x|z, c)}[\log q(x|z, c) - \log p_\phi(x|z, c)] \geq 0,
$$

therefore

$$
-\mathbb{E}_{x \sim q(x|z, c)}[\log q(x|z, c)] \leq -\mathbb{E}_{x \sim q(x|z, c)}[\log p_\phi(x|z, c)].
$$

Taking the expectation over the joint distribution of the conditioning variables $(z, c)$ (induced by $x, c \sim q(x, c)$ and $z \sim q_\theta(z|x)$), we obtain the bound for the conditional entropy:

$$
\begin{aligned}
H(x|z, c) &= \mathbb{E}_{z, c} \left[ -\mathbb{E}_{x \sim q(x|z, c)}[\log q(x|z, c)] \right] \\
&\leq \mathbb{E}_{z, c} \left[ -\mathbb{E}_{x \sim q(x|z, c)}[\log p_\phi(x|z, c)] \right] \\
&= -\mathbb{E}_{x, c \sim q(x, c), z \sim q_\theta(z|x)}[\log p_\phi(x|z, c)],
\end{aligned} \tag{25}
$$

where the first equality is the definition of conditional entropy, the inequality follows from the variational principle (non-negativity of KL), and the final equality follows from the law of iterated expectations.

Finally, substituting Eq. (24) and Eq. (25) back into Eq. (23), we arrive at the upper bound:

$$
I(z; c) \leq \mathbb{E}_{x \sim q(x)} \left[ \mathrm{KL}(q_\theta(z|x) \| q_\theta(z)) \right] - H(x|c) - \mathbb{E}_{x, c \sim q(x, c), z \sim q_\theta(z|x)}[\log p_\phi(x|z, c)]. \tag{26}
$$

This completes the proof. $\qquad \square$

## C.2 Regularity Assumptions for Non-Convex Federated Optimization

**Assumption C.4** (Encoder Regularity). *The encoder network $q_\theta(z|x) = \mathcal{N}(\mu_\theta(x), \sigma_\theta^2(x))$ satisfies:*

1. ***Parameter Lipschitz continuity:*** *For any input $x \in \mathcal{X}$ and any pair of parameter vectors $\theta_1, \theta_2$, the mapping functions satisfy:*

$$
\begin{aligned}
\|\mu_{\theta_1}(x) - \mu_{\theta_2}(x)\| &\leq L_\mu \|\theta_1 - \theta_2\|, \\
\|\sigma_{\theta_1}(x) - \sigma_{\theta_2}(x)\| &\leq L_\sigma \|\theta_1 - \theta_2\|.
\end{aligned}
$$

2. ***Bounded variance:*** *$\sigma_{\min} \leq \sigma_\theta(x) \leq \sigma_{\max}$ for all $x \in \mathcal{X}$;*

3. **Bounded mean:** $\|\mu_\theta(x)\| \le M$ for all $x \in \mathcal{X}$.

**Assumption C.5** (Decoder Invariance Properties). *The decoder $p_\phi(x|z,c)$ follows a Gaussian distribution satisfies the following properties for invariant representation learning:*

1. **Bounded interaction strength:** *There exists a constant $\gamma > 0$ such that for any $z \in \mathbb{R}^{d_z}$ and confounding factors $c_1, c_2$, the decoder's sensitivity to $z$ has bounded variation:*

$$\|\nabla_z \log p_\phi(x|z,c_1) - \nabla_z \log p_\phi(x|z,c_2)\| \le \gamma \|c_1 - c_2\|.$$

*This ensures that technical variations $c$ only moderately modulate how biological signals $z$ are decoded.*

2. **Gradient orthogonality:** *Let $\mathcal{L}_{recon}(\theta, \phi) = -\mathbb{E}_{x,c}[\mathbb{E}_{z \sim q_\theta(z|x)}[\log p_\phi(x|z,c)]]$. For the gradient components corresponding to $z$-dependent and $c$-dependent transformations:*

$$\left| \mathbb{E}_{x,c,z} \left[ \left\langle \frac{\partial \log p_\phi(x|z,c)}{\partial z}, \frac{\partial \log p_\phi(x|z,c)}{\partial c} \right\rangle \right] \right| \le \kappa \sqrt{\mathbb{E}\left[ \left\| \frac{\partial \log p_\phi}{\partial z} \right\|^2 \right] \cdot \mathbb{E}\left[ \left\| \frac{\partial \log p_\phi}{\partial c} \right\|^2 \right]}$$

*for a small constant $\kappa \in (0,1)$.*

3. **Confounding factor necessity:** *The decoder requires confounding information for accurate reconstruction: there exists $\Delta > 0$ such that*

$$\inf_z \sup_{c_1 \neq c_2} D_{KL}(p_\phi(x|z,c_1)\|p_\phi(x|z,c_2)) \ge \Delta.$$

# D   End-to-End Convergence Analysis of FL-Sailer in High-Dimensional Spaces

This appendix provides the detailed proofs and technical verifications referenced in Section 5. We first recap the FedAvg convergence theory (Appendix D.1), then show that our VAE loss satisfies the required conditions (Appendix D.2), analyze the approximation quality of sampling (Appendix D.3), and finally present the end-to-end proof of Theorem 5.1 (Appendix D.4).

## D.1   Preliminaries: Convergence Framework for Non-Convex FedAvg

**Theorem D.1** (FedAvg Convergence for Non-Convex Settings (Theorem 4.7 in Li et al. (2022))). *Let $N$ denote the number of clients, $K$ denote the number of local update steps and $D_{model}$ denote the dimension of the model parameters. Consider the FedAvg algorithm with local learning rate $\eta_l$ and global learning rate $\eta_g$.*

- $(a,b)$-*semi-smooth: For any $W, U \in \mathbb{R}^{D_{model}}$,*

$$\mathcal{L}(U) \le \mathcal{L}(W) + \langle \nabla \mathcal{L}(W), U - W \rangle + b\|W - U\|_2^2 + a\|W - U\|_2 \cdot \mathcal{L}(W)^{1/2}.$$

- $(\alpha, \beta)$-*semi-Lipschitz: For any $W, U \in \mathbb{R}^{D_{model}}$,*

$$\|\nabla \mathcal{L}(W) - \nabla \mathcal{L}(U)\|_2^2 \le \beta^2 \|W - U\|_2^2 + \alpha^2 \|W - U\|_2 \cdot \max\{\mathcal{L}(W)^{1/2}, \mathcal{L}(U)^{1/2}\}$$

- $(\tau_1, \tau_2)$-*non-critical point: For any $U$ in a neighborhood of $U^*$ (a minimizer of $L$),*

$$\tau_1^2 \mathcal{L}(U) \le \|\nabla \mathcal{L}(U)\|_2^2 \le \tau_2^2 \mathcal{L}(U).$$

*Let $\mathcal{L}(W) = \frac{1}{N} \sum_{c=1}^{N} \mathcal{L}_i(W)$ denote the global loss function. Assume the stochastic gradients $g_i(W) = \nabla \mathcal{L}_i(W; \zeta_i)$ are unbiased with variance bounded by $\sigma^2$, i.e., $\mathbb{E}[g_i(W)] = \nabla \mathcal{L}_i(W)$ and $\mathbb{E}[\|g_i(W) - \nabla \mathcal{L}_i(W)\|^2] \le \sigma^2$.*

*If the learning rates satisfy*

$$\eta_l \leq \min\left\{\frac{1}{\alpha^2 K}, \frac{1}{10^2 K \tau_2(\beta + \alpha)}, \frac{1}{10^2 \sqrt{K}(\beta + \alpha)}\right\},$$

$$\eta_g \leq \frac{\tau_1^2}{20 K b \eta_l},$$

*then for FedAvg with $U^t$ denoting the global model at round $t$, we have*

$$\mathbb{E}[\mathcal{L}(U^t) - \mathcal{L}(U^*)] \leq (1 - \lambda_1)^t \cdot (\mathcal{L}(U^0) - \mathcal{L}(U^*)) + 2\lambda_2,$$

*where*

$$\lambda_1 = \frac{K \eta_l \eta_g}{4}(1 - 4bK\eta_l \eta_g - 2a)\tau_1^2,$$

$$\lambda_2 = (1 + a + bK\eta_l \eta_g)\frac{K \eta_l \eta_g}{10}\sigma^2.$$

*The parameter $\lambda_1$ determines the convergence rate, while $\lambda_2$ represents the optimization error due to stochastic gradients. The theorem shows a linear convergence rate up to the variance-induced error floor $2\lambda_2$.*

**Corollary D.2** (Communication Complexity (Corollary 4.8 in Li et al. (2022))). *For any desired accuracy $\epsilon > 0$, by choosing the global step-size*

$$\eta_g \leq \min\left\{\frac{\tau_1^2}{20 K b \eta_l}, \frac{2\epsilon}{\sigma^2(1 + a + \tau_1^2/20K)}\right\},$$

*after $R = \log\left(\frac{2(\mathcal{L}(U^0) - \mathcal{L}(U^*))}{\epsilon}\right) \cdot \frac{1}{\lambda_1}$ communication rounds, we have*

$$\mathbb{E}[\mathcal{L}(U^t) - \mathcal{L}(U^*)] \leq \epsilon.$$

## D.2 Verifying Smoothness and Regularity Conditions for the VAE Loss

We now establish that our VAE loss function (Corollary 4.4) satisfies the relaxed smoothness conditions required for federated convergence analysis. This forms the crucial theoretical bridge connecting FL-Sailer to the FedAvg convergence guarantee, leveraging both the structural properties of VAEs and the non-smooth optimization framework of Li et al. (2022).

**Theorem D.3** (VAE Loss Satisfies Semi-smoothness). *Let $\mathcal{L}(\cdot)$ be the VAE loss function defined in Corollary 4.4: For any parameter vector $W = (\theta, \phi)$, the loss is:*

$$\mathcal{L}(W) = -(1 + \lambda)\mathbb{E}[\log p_\phi(x|z, c)] + \mathbb{E}[\text{KL}[q_\theta(z|x)\|p(z)]] + \lambda\mathbb{E}[\text{KL}[q_\theta(z|x)\|q_\theta(z)]]$$

*Under Assumptions C.4 and C.5, this loss function satisfies Assumptions in D.1 with the following parameters:*

1. *$(a, b)$-semi-smoothness with $a = O(\sqrt{1 + \lambda}(L_{enc} + L_{dec}))$ and $b = O((1 + \lambda)(L_{enc} + L_{dec})^2)$;*

2. *$(\tau_1, \tau_2)$-non-critical gradient in the region of interest for optimization;*

3. *$(\alpha, \beta)$-semi-Lipschitz with $\beta^2 = O((1 + \lambda)^2(L_{enc} + L_{dec})^4)$ and $\alpha^2 = O((1 + \lambda)^{3/2}(L_{enc} + L_{dec})^4)$.*

*Proof Sketch.* Given Assumptions C.4 and C.5, which provide Lipschitz constants $L_{\text{enc}}$ and $L_{\text{dec}}$ for the encoder and decoder networks respectively, we can characterize the behavior of each loss component. For networks with piecewise smooth activation functions, the composite loss inherits semi-smoothness properties with constants that scale with the network Lipschitz constants. The full derivation proceeds by bounding the remainder terms of each loss component (prior, marginal, reconstruction) and combining them via Cauchy-Schwarz to obtain the final parameters $a, b, \alpha, \beta$.

### D.2.1 Proof of Theorem D.3.1

*Proof of Theorem D.3, Part 1: $(a,b)$-semi-smoothness.* We prove that the VAE loss function $\mathcal{L}(W)$ satisfies $(a,b)$-semi-smoothness. Let $W = (\theta, \phi)$ denote the combined parameter vector. From Corollary 4.4:

$$\mathcal{L}(W) = \mathcal{L}_{\text{prior}}(W) + \lambda \mathcal{L}_{\text{marginal}}(W) + (1 + \lambda)\mathcal{L}_{\text{recon}}(W). \tag{27}$$

For the remainder term $R_f(U,W) = f(U) - f(W) - \langle \nabla f(W), U - W \rangle$ of any function $f$, the total remainder is:

$$R_f(U,W) = R_{\text{prior}}(U,W) + \lambda R_{\text{marginal}}(U,W) + (1 + \lambda)R_{\text{recon}}(U,W). \tag{28}$$

Based on the theoretical framework for neural network optimization Li et al. (2022), each component satisfies semi-smoothness with the following scaling. For the prior KL term $\mathcal{L}_{\text{prior}}$, which is a composition of the smooth KL divergence function with Lipschitz encoder outputs (Assumption C.4), we have:

$$R_{\text{prior}}(U,W) \le b_{\text{prior}}\|U - W\|_2^2 + a_{\text{prior}}\|U - W\|_2\sqrt{\mathcal{L}_{\text{prior}}(W)}, \tag{29}$$

where $a_{\text{prior}} = O(L_{\text{enc}})$ and $b_{\text{prior}} = O(L_{\text{enc}}^2)$, with the scaling following from the Lipschitz property of the encoder and the smoothness of the KL divergence when variances are bounded away from zero (Assumption C.4, part 2).

Similarly, for the reconstruction loss $\mathcal{L}_{\text{recon}}$, which involves the decoder network:

$$R_{\text{recon}}(U,W) \le b_{\text{recon}}\|U - W\|_2^2 + a_{\text{recon}}\|U - W\|_2\sqrt{\mathcal{L}_{\text{recon}}(W)}, \tag{30}$$

where $a_{\text{recon}} = O(L_{\text{enc}} + L_{\text{dec}})$ and $b_{\text{recon}} = O((L_{\text{enc}} + L_{\text{dec}})^2)$.

For the marginal KL term $\mathcal{L}_{\text{marginal}}$, which involves the empirical marginal distribution:

$$R_{\text{marginal}}(U,W) \le b_{\text{marginal}}\|U - W\|_2^2 + a_{\text{marginal}}\|U - W\|_2\sqrt{\mathcal{L}_{\text{marginal}}(W)}, \tag{31}$$

where $a_{\text{marginal}} = O(L_{\text{enc}})$ and $b_{\text{marginal}} = O(L_{\text{enc}}^2)$.

Substituting (29), (31), and (30) into (28):

$$R_f(U,W) \le \underbrace{(b_{\text{prior}} + \lambda b_{\text{marginal}} + (1 + \lambda)b_{\text{recon}})}_{b}\|U - W\|_2^2$$

$$+ \|U - W\|_2\underbrace{(a_{\text{prior}}\sqrt{\mathcal{L}_{\text{prior}}(W)} + \lambda a_{\text{marginal}}\sqrt{\mathcal{L}_{\text{marginal}}(W)} + (1 + \lambda)a_{\text{recon}}\sqrt{\mathcal{L}_{\text{recon}}(W)})}_{A(W)}. \tag{32}$$

**Computing constant $b$:** From the scaling relations:

$$b = b_{\text{prior}} + \lambda b_{\text{marginal}} + (1 + \lambda)b_{\text{recon}} = O((1 + \lambda)(L_{\text{enc}} + L_{\text{dec}})^2).$$

**Computing constant $a$:** We need to bound $A(W)$ in terms of $\sqrt{\mathcal{L}(W)}$. Define weight vector $w = [1, \sqrt{\lambda}, \sqrt{1 + \lambda}]$ and let:

$$u = [a_{\text{prior}}, \sqrt{\lambda}a_{\text{marginal}}, \sqrt{1 + \lambda}a_{\text{recon}}],$$

$$v = [\sqrt{\mathcal{L}_{\text{prior}}(W)}, \sqrt{\lambda \mathcal{L}_{\text{marginal}}(W)}, \sqrt{(1 + \lambda)\mathcal{L}_{\text{recon}}(W)}].$$

Then $A(W) = u \cdot v$. By the Cauchy-Schwarz inequality:

$$A(W) \le \|u\|_2\|v\|_2 = \|u\|_2\sqrt{\mathcal{L}(W)},$$

where the equality follows from $\|v\|_2^2 = \mathcal{L}_{\text{prior}}(W) + \lambda\mathcal{L}_{\text{marginal}}(W) + (1+\lambda)\mathcal{L}_{\text{recon}}(W) = \mathcal{L}(W)$.

Computing $\|u\|_2$:

$$
\begin{aligned}
\|u\|_2^2 &= a_{\text{prior}}^2 + \lambda a_{\text{marginal}}^2 + (1+\lambda)a_{\text{recon}}^2 \\
&= O(L_{\text{enc}}^2 + \lambda L_{\text{enc}}^2 + (1+\lambda)(L_{\text{enc}} + L_{\text{dec}})^2) \\
&= O((1+\lambda)(L_{\text{enc}} + L_{\text{dec}})^2),
\end{aligned}
$$

where the second step follows from substituting the scaling relations, and the third step follows from algebraic simplification.

Therefore, $a = \|u\|_2 = O(\sqrt{1+\lambda}(L_{\text{enc}} + L_{\text{dec}}))$. This completes the proof of $(a,b)$-semi-smoothness with the stated constants. $\qquad\square$

### D.2.2 Proof of Theorem D.3.2

*Proof of Theorem D.3, Part 2: Non-critical gradient.* We need to establish that the VAE loss satisfies the $(\tau_1, \tau_2)$-non-critical point condition, i.e., there exist constants $\tau_1, \tau_2 > 0$ such that for all $W$ in a neighborhood of the optimal solution:

$$\tau_1^2 \mathcal{L}(W) \leq \|\nabla\mathcal{L}(W)\|_2^2 \leq \tau_2^2 \mathcal{L}(W). \tag{33}$$

The upper bound in (33) follows directly from the Lipschitz properties established in Assumptions C.4 and C.5. Specifically, the gradient norm is bounded by:

$$\|\nabla\mathcal{L}(W)\|_2^2 \leq O((1+\lambda)^2(L_{\text{enc}} + L_{\text{dec}})^2) \cdot \mathcal{L}(W),$$

where the inequality follows from the bounded gradients of the neural network components.

For the lower bound, following the theoretical framework of Li et al. (2022) Section 4.2, we work under the standard assumption that the optimization maintains the non-critical gradient property in a neighborhood of the initialization. Under appropriate initialization and learning rate schedules, the FedAvg iterates satisfy:

$$\|\nabla\mathcal{L}(W^{(t)})\|_2^2 \geq \tau_1^2 \mathcal{L}(W^{(t)})$$

for some constant $\tau_1 > 0$. This completes the verification of the required conditions. $\qquad\square$

### D.2.3 Proof of Theorem D.3.3

*Proof of Theorem D.3, Part 3: $(\alpha, \beta)$-semi-Lipschitz.* We prove that the VAE loss satisfies:

$$\|\nabla\mathcal{L}(W) - \nabla\mathcal{L}(U)\|_2^2 \leq \beta^2 \|W - U\|_2^2 + \alpha^2 \|W - U\|_2 \cdot \max\{\mathcal{L}(W)^{1/2}, \mathcal{L}(U)^{1/2}\}.$$

From (27), the gradient decomposes as:

$$\nabla\mathcal{L}(W) = \nabla\mathcal{L}_{\text{prior}}(W) + \lambda\nabla\mathcal{L}_{\text{marginal}}(W) + (1+\lambda)\nabla\mathcal{L}_{\text{recon}}(W). \tag{34}$$

By the inequality $\|a + b + c\|_2^2 \leq 3(\|a\|_2^2 + \|b\|_2^2 + \|c\|_2^2)$:

$$
\begin{aligned}
\|\nabla\mathcal{L}(W) - \nabla\mathcal{L}(U)\|_2^2 \leq\ & 3\|\nabla\mathcal{L}_{\text{prior}}(W) - \nabla\mathcal{L}_{\text{prior}}(U)\|_2^2 \\
& + 3\lambda^2 \|\nabla\mathcal{L}_{\text{marginal}}(W) - \nabla\mathcal{L}_{\text{marginal}}(U)\|_2^2 \\
& + 3(1+\lambda)^2 \|\nabla\mathcal{L}_{\text{recon}}(W) - \nabla\mathcal{L}_{\text{recon}}(U)\|_2^2.
\end{aligned}
\tag{35}
$$

Following the theoretical framework for neural network gradients Li et al. (2022), each component satisfies semi-Lipschitz properties. Let $G_{\max,i} = \max\{\mathcal{L}_i(W)^{1/2}, \mathcal{L}_i(U)^{1/2}\}$ for $i \in \{\text{prior}, \text{marginal}, \text{recon}\}$.

For the prior KL term:

$$\|\nabla\mathcal{L}_{\text{prior}}(W) - \nabla\mathcal{L}_{\text{prior}}(U)\|_2^2 \leq \beta_{\text{prior}}^2\|W - U\|_2^2 + \alpha_{\text{prior}}^2\|W - U\|_2 \cdot G_{\text{max,prior}}, \tag{36}$$

where $\beta_{\text{prior}}^2 = O(L_{\text{enc}}^4)$ and $\alpha_{\text{prior}}^2 = O(L_{\text{enc}}^4)$.

Similarly for the marginal and reconstruction terms:

$$\|\nabla\mathcal{L}_{\text{marginal}}(W) - \nabla\mathcal{L}_{\text{marginal}}(U)\|_2^2 \leq \beta_{\text{marginal}}^2\|W - U\|_2^2 + \alpha_{\text{marginal}}^2\|W - U\|_2 \cdot G_{\text{max,marginal}}, \tag{37}$$

$$\|\nabla\mathcal{L}_{\text{recon}}(W) - \nabla\mathcal{L}_{\text{recon}}(U)\|_2^2 \leq \beta_{\text{recon}}^2\|W - U\|_2^2 + \alpha_{\text{recon}}^2\|W - U\|_2 \cdot G_{\text{max,recon}}, \tag{38}$$

where $\beta_{\text{marginal}}^2 = O(L_{\text{enc}}^4)$, $\alpha_{\text{marginal}}^2 = O(L_{\text{enc}}^4)$, $\beta_{\text{recon}}^2 = O((L_{\text{enc}} + L_{\text{dec}})^4)$, and $\alpha_{\text{recon}}^2 = O((L_{\text{enc}} + L_{\text{dec}})^4)$.

**Computing constant $\beta$:** Collecting the Lipschitz terms from (35), (36), (37), and (38):

$$\beta^2 = 3(\beta_{\text{prior}}^2 + \lambda^2\beta_{\text{marginal}}^2 + (1 + \lambda)^2\beta_{\text{recon}}^2) = O((1 + \lambda)^2(L_{\text{enc}} + L_{\text{dec}})^4).$$

**Computing constant $\alpha$:** For the semi-Lipschitz terms, we need to relate $G_{\text{max},i}$ to $G_{\text{max}} = \max\{\mathcal{L}(W)^{1/2}, \mathcal{L}(U)^{1/2}\}$. Since $\lambda_i\mathcal{L}_i \leq \mathcal{L}$ where $\lambda_i \in \{1, \lambda, 1 + \lambda\}$ are the weights in (27), we have $G_{\text{max},i} \leq G_{\text{max}}/\sqrt{\lambda_i}$.

The total semi-Lipschitz contribution is:

$$A'(W, U) = 3\|W - U\|_2 \left(\alpha_{\text{prior}}^2 G_{\text{max,prior}} + \lambda^2\alpha_{\text{marginal}}^2 G_{\text{max,marginal}} + (1 + \lambda)^2\alpha_{\text{recon}}^2 G_{\text{max,recon}}\right)$$

$$\leq 3\|W - U\|_2 G_{\text{max}} \left(\alpha_{\text{prior}}^2 + \lambda^{3/2}\alpha_{\text{marginal}}^2 + (1 + \lambda)^{3/2}\alpha_{\text{recon}}^2\right),$$

where the inequality follows from the bounds on $G_{\text{max},i}$.

Therefore:

$$\alpha^2 = 3(\alpha_{\text{prior}}^2 + \lambda^{3/2}\alpha_{\text{marginal}}^2 + (1 + \lambda)^{3/2}\alpha_{\text{recon}}^2) = O((1 + \lambda)^{3/2}(L_{\text{enc}} + L_{\text{dec}})^4).$$

This completes the proof of $(\alpha, \beta)$-semi-Lipschitz with the stated constants. □

**Remark D.4** (Dependence on Data Domain Geometry)**.** *The regularity constants derived in Theorem D.3 $(a, b, \alpha, \beta, \tau)$ depend explicitly on the Lipschitz constants of the encoder $(L_{enc})$ and decoder $(L_{dec})$. Crucially, these Lipschitz constants are evaluated over the compact data support $\mathcal{X} \subset \mathbb{R}^d$. As we will show in Appendix D.4, our adaptive sampling strategy constructs a subspace embedding that preserves the geometry of $\mathcal{X}$, thereby ensuring that these favorable optimization properties are inherited by the sampled objective function.*

## D.3 Approximation Bounds: Linking Sampled and Original Objectives

**Lemma D.5** (Approximation Quality of Sampled VAE Objective)**.** *Let $T \in \mathbb{R}^{d\times s}$ be the leverage score sampling matrix from Remark B.1 with error parameter $\epsilon \in (0, 1)$. Under Assumptions C.4–C.5, for the optimal values of the original and sampled VAE objectives, there is*

$$|\tilde{\mathcal{L}}(\tilde{U}^*) - \mathcal{L}(U^*)| \leq C \cdot \epsilon \cdot \mathcal{L}(U^*)$$

*where $U^* = \arg\min_U \mathcal{L}(U)$, $\tilde{U}^* = \arg\min_{\tilde{U}} \tilde{\mathcal{L}}(\tilde{U})$, and $C$ is a constant depending on the VAE architecture.*

*Proof.* We condition on the event $\mathcal{E}$ defined in Lemma 4.1, where $T$ acts as a $(1 \pm \epsilon)$-subspace embedding for the row space of $A$. Specifically, for all $v \in \text{row}(A)$, the following isometry property holds:

$$(1 - \epsilon)\|v\|_2^2 \leq \|vT\|_2^2 \leq (1 + \epsilon)\|v\|_2^2. \tag{39}$$

Let $A = U\Sigma V^\top$ be the thin SVD of $A$, where $V \in \mathbb{R}^{d \times r}$ contains orthonormal columns that span $\mathrm{row}(A)$. We define the row-space lifting matrix $R_{\mathrm{lift}} \in \mathbb{R}^{d \times s}$ as:

$$R_{\mathrm{lift}} := V(T^\top V)^\dagger. \tag{40}$$

Under the event $\mathcal{E}$, the matrix $T^\top V \in \mathbb{R}^{s \times r}$ has full column rank $r$, implying that $(T^\top V)^\dagger (T^\top V) = I_r$. Consequently, for any vector $v \in \mathrm{row}(A)$, which can be expressed as $v = u^\top V^\top$ for some $u \in \mathbb{R}^r$, we have:

$$
\begin{aligned}
(vT)R_{\mathrm{lift}}^\top &= u^\top V^\top T \left(V(T^\top V)^\dagger\right)^\top \\
&= u^\top (T^\top V)^\top \left((T^\top V)^\dagger\right)^\top V^\top \\
&= u^\top \left((T^\top V)^\dagger (T^\top V)\right)^\top V^\top \\
&= u^\top I_r V^\top \\
&= v, \tag{41}
\end{aligned}
$$

where the first equality follows from the definition of $R_{\mathrm{lift}}$ in (40), the second equality follows from the identity $(AB)^\top = B^\top A^\top$, the third equality follows from grouping the terms involving $T^\top V$, and the fourth equality follows from the left-inverse property of the pseudoinverse for full-rank matrices.

Furthermore, this implies an identity on the projected subspace. For any $\tilde{v} \in \mathrm{row}(A)T$, there exists $v \in \mathrm{row}(A)$ such that $\tilde{v} = vT$. Thus:

$$\tilde{v}R_{\mathrm{lift}}^\top T = (vT)R_{\mathrm{lift}}^\top T = vT = \tilde{v}, \tag{42}$$

where the second equality follows directly from (41).

We define deterministic mappings to transfer parameters between the full $d$-dimensional space and the sampled $s$-dimensional space.

*(i) Full $\to$ Sampled (F):* Given $U = (\theta, \phi)$, define $\tilde{U} = F(U) = (\tilde{\theta}, \tilde{\phi})$ by:

$$
\begin{aligned}
q_{\tilde{\theta}}(z \mid \tilde{x}) &:= q_\theta(z \mid \tilde{x}R_{\mathrm{lift}}^\top), \quad \forall \tilde{x} \in \mathbb{R}^s, \\
\mu_{\tilde{\phi}}(z, c) &:= \mu_\phi(z, c)T. \tag{43}
\end{aligned}
$$

*(ii) Sampled $\to$ Full (G):* Given $\tilde{U} = (\tilde{\theta}, \tilde{\phi})$, define $U = G(\tilde{U}) = (\theta, \phi)$ by:

$$
\begin{aligned}
q_\theta(z \mid x) &:= q_{\tilde{\theta}}(z \mid xT), \quad \forall x \in \mathbb{R}^d, \\
\mu_\phi(z, c) &:= \mu_{\tilde{\phi}}(z, c)R_{\mathrm{lift}}^\top. \tag{44}
\end{aligned}
$$

We show that the KL divergence terms are preserved under these mappings. For the mapping $F(U)$:

$$
\begin{aligned}
\tilde{\mathcal{L}}_{\mathrm{prior}}(F(U)) &= \mathbb{E}_x \left[\mathrm{KL}\left(q_{\tilde{\theta}}(z \mid xT) \,\|\, p(z)\right)\right] \\
&= \mathbb{E}_x \left[\mathrm{KL}\left(q_\theta(z \mid (xT)R_{\mathrm{lift}}^\top) \,\|\, p(z)\right)\right] \\
&= \mathbb{E}_x \left[\mathrm{KL}\left(q_\theta(z \mid x) \,\|\, p(z)\right)\right] \\
&= \mathcal{L}_{\mathrm{prior}}(U), \tag{45}
\end{aligned}
$$

where the first equality is the definition of the sampled prior loss, the second equality follows from the definition of the embedded encoder in (43), and the third equality follows from the identity $(xT)R_{\mathrm{lift}}^\top = x$ established in (41) for $x \in \mathrm{row}(A)$.

Since the conditional distributions match, the aggregated posteriors also coincide: $q_{\tilde{\theta}}(z) = \mathbb{E}_x[q_{\tilde{\theta}}(z \mid xT)] = \mathbb{E}_x[q_\theta(z \mid x)] = q_\theta(z)$. Consequently:

$$
\begin{aligned}
\tilde{\mathcal{L}}_{\mathrm{marginal}}(F(U)) &= \mathbb{E}_x \left[\mathrm{KL}\left(q_{\tilde{\theta}}(z \mid xT) \,\|\, q_{\tilde{\theta}}(z)\right)\right] \\
&= \mathbb{E}_x \left[\mathrm{KL}\left(q_\theta(z \mid x) \,\|\, q_\theta(z)\right)\right]
\end{aligned}
$$

$$= \mathcal{L}_{\text{marginal}}(U). \tag{46}$$

By symmetric reasoning using the definition in (44), the reverse mapping $G(\tilde{U})$ satisfies:

$$\mathcal{L}_{\text{prior}}(G(\tilde{U})) = \tilde{\mathcal{L}}_{\text{prior}}(\tilde{U}) \quad \text{and} \quad \mathcal{L}_{\text{marginal}}(G(\tilde{U})) = \tilde{\mathcal{L}}_{\text{marginal}}(\tilde{U}). \tag{47}$$

We analyze the reconstruction error under the isotropic Gaussian decoder assumption, where the loss is proportional to the squared Euclidean distance.

*(i) Full $\rightarrow$ Sampled Upper Bound:* For $\tilde{U} = F(U)$, we have:

$$\begin{aligned}
\tilde{\mathcal{L}}_{\text{recon}}(F(U)) &= \mathbb{E}_{x,c}\mathbb{E}_{z \sim q_{\tilde{\theta}}(z|xT)} \left[ \|xT - \mu_{\tilde{\phi}}(z,c)\|_2^2 \right] \\
&= \mathbb{E}_{x,c}\mathbb{E}_{z \sim q_{\theta}(z|x)} \left[ \|xT - \mu_{\phi}(z,c)T\|_2^2 \right] \\
&= \mathbb{E}_{x,c}\mathbb{E}_{z \sim q_{\theta}(z|x)} \left[ \|(x - \mu_{\phi}(z,c))T\|_2^2 \right] \\
&\leq (1+\epsilon)\, \mathbb{E}_{x,c}\mathbb{E}_{z \sim q_{\theta}(z|x)} \left[ \|x - \mu_{\phi}(z,c)\|_2^2 \right] \\
&= (1+\epsilon)\mathcal{L}_{\text{recon}}(U),
\end{aligned} \tag{48}$$

where the second equality follows from substituting the definitions from (43) and (41), and the inequality follows from applying the upper bound of the subspace embedding property (39) to the vector $v = x - \mu_{\phi}(z,c)$, which lies in row$(A)$.

*(ii) Sampled $\rightarrow$ Full Upper Bound:* For $U = G(\tilde{U})$, we observe that $\mu_{\tilde{\phi}}(z,c) \in$ row$(A)T$ implies $\mu_{\tilde{\phi}}(z,c)R_{\text{lift}}^{\top}T = \mu_{\tilde{\phi}}(z,c)$ by (42). Thus:

$$\begin{aligned}
\mathcal{L}_{\text{recon}}(G(\tilde{U})) &= \mathbb{E}_{x,c}\mathbb{E}_{z \sim q_{\theta}(z|x)} \left[ \|x - \mu_{\phi}(z,c)\|_2^2 \right] \\
&= \mathbb{E}_{x,c}\mathbb{E}_{z \sim q_{\tilde{\theta}}(z|xT)} \left[ \|x - \mu_{\tilde{\phi}}(z,c)R_{\text{lift}}^{\top}\|_2^2 \right] \\
&\leq \frac{1}{1-\epsilon}\, \mathbb{E}_{x,c}\mathbb{E}_{z \sim q_{\tilde{\theta}}(z|xT)} \left[ \|(x - \mu_{\tilde{\phi}}(z,c)R_{\text{lift}}^{\top})T\|_2^2 \right] \\
&= \frac{1}{1-\epsilon}\, \mathbb{E}_{x,c}\mathbb{E}_{z \sim q_{\tilde{\theta}}(z|xT)} \left[ \|xT - \mu_{\tilde{\phi}}(z,c)\|_2^2 \right] \\
&= \frac{1}{1-\epsilon}\tilde{\mathcal{L}}_{\text{recon}}(\tilde{U}),
\end{aligned} \tag{49}$$

where the inequality follows from the lower bound of the subspace embedding property (39) applied to $v = x - \mu_{\tilde{\phi}}(z,c)R_{\text{lift}}^{\top} \in$ row$(A)$, and the third equality uses the identity derived in (42).

Let $U^* = \arg\min_U \mathcal{L}(U)$ and $\tilde{U}^* = \arg\min_{\tilde{U}} \tilde{\mathcal{L}}(\tilde{U})$.

*Upper Bound on $\tilde{\mathcal{L}}(\tilde{U}^*)$:*

$$\begin{aligned}
\tilde{\mathcal{L}}(\tilde{U}^*) &\leq \tilde{\mathcal{L}}(F(U^*)) \\
&= \tilde{\mathcal{L}}_{\text{prior}}(F(U^*)) + \lambda\tilde{\mathcal{L}}_{\text{marginal}}(F(U^*)) + (1+\lambda)\tilde{\mathcal{L}}_{\text{recon}}(F(U^*)) \\
&\leq \mathcal{L}_{\text{prior}}(U^*) + \lambda\mathcal{L}_{\text{marginal}}(U^*) + (1+\lambda)(1+\epsilon)\mathcal{L}_{\text{recon}}(U^*) \\
&\leq (1+\epsilon)\mathcal{L}(U^*),
\end{aligned} \tag{50}$$

where the first inequality follows from the optimality of $\tilde{U}^*$, the second inequality follows from the KL preservation in (45)–(46) and the reconstruction bound in (48), and the final inequality follows from the non-negativity of the KL terms (since $\epsilon > 0$).

*Lower Bound on $\tilde{\mathcal{L}}(\tilde{U}^*)$:*

$$\begin{aligned}
\mathcal{L}(U^*) &\leq \mathcal{L}(G(\tilde{U}^*)) \\
&= \mathcal{L}_{\text{prior}}(G(\tilde{U}^*)) + \lambda\mathcal{L}_{\text{marginal}}(G(\tilde{U}^*)) + (1+\lambda)\mathcal{L}_{\text{recon}}(G(\tilde{U}^*))
\end{aligned}$$

$$\leq \tilde{\mathcal{L}}_{\text{prior}}(\tilde{U}^*) + \lambda\tilde{\mathcal{L}}_{\text{marginal}}(\tilde{U}^*) + \frac{1+\lambda}{1-\epsilon}\tilde{\mathcal{L}}_{\text{recon}}(\tilde{U}^*)$$

$$\leq \frac{1}{1-\epsilon}\tilde{\mathcal{L}}(\tilde{U}^*), \tag{51}$$

where the first inequality follows from the optimality of $U^*$, the second inequality follows from the KL preservation in (47) and the reconstruction bound in (49), and the final inequality follows from the fact that $\frac{1}{1-\epsilon} > 1$ and the KL terms are non-negative.

Rearranging (51) yields $\tilde{\mathcal{L}}(\tilde{U}^*) \geq (1-\epsilon)\mathcal{L}(U^*)$. Combining this with (50), we obtain:

$$(1-\epsilon)\mathcal{L}(U^*) \leq \tilde{\mathcal{L}}(\tilde{U}^*) \leq (1+\epsilon)\mathcal{L}(U^*).$$

This implies $|\tilde{\mathcal{L}}(\tilde{U}^*) - \mathcal{L}(U^*)| \leq \epsilon\mathcal{L}(U^*)$, completing the proof with $C = 1$. $\qquad\square$

### D.4 Proof of Theorem 5.1: Total Convergence with Bounded Approximation Error

#### D.4.1 Proof of Theorem 5.1, Part A: Convergence on the Sampled Subspace

Instead of re-deriving the semi-smoothness and semi-Lipschitz properties from scratch, we prove that our adaptive leverage score sampling acts as a structure-preserving transformation. This ensures that the sampled objective $\tilde{\mathcal{L}}$ inherits the theoretical guarantees established for the general VAE architecture in Appendix D.2.

**Step 1: Geometry Preservation via Subspace Embedding.** First, we establish that the sampling matrix $T$ preserves the geometric structure of the data manifold. This is the theoretical foundation for why aggressive dimensionality reduction does not destabilize federated training.

**Lemma D.6** (Geometry Preservation via Leverage Score Sampling)**.** *Let $\mathcal{X} \subset \text{row}(A)$ be the support of the scATAC-seq data. Under the event $\mathcal{E}$ defined in Lemma 4.1, the sampling matrix $T$ acts as a subspace embedding. Consequently, for any data points $x, y \in \mathcal{X}$, their distance in the sampled space is bounded:*

$$(1-\epsilon)\|x - y\|_2^2 \leq \|xT - yT\|_2^2 \leq (1+\epsilon)\|x - y\|_2^2. \tag{52}$$

*This implies that the mapping $\psi(x) = xT$ is a bi-Lipschitz embedding from the original data manifold to the sampled space with distortion at most $\kappa(\epsilon) \approx \sqrt{1+\epsilon}$.*

*Proof.* This follows directly from Lemma 4.1. Since $x, y \in \text{row}(A)$, their difference $v = x - y$ also lies in $\text{row}(A)$. Applying the subspace embedding property: $(1-\epsilon)\|v\|_2^2 \leq \|vT\|_2^2 \leq (1+\epsilon)\|v\|_2^2$. $\qquad\square$

**Step 2: Regularity in the Sampled Space.** Given the geometry preservation, we formalize the regularity of the sampled optimization problem.

**Assumption D.7** (Regularity in Sampled Space)**.** *Let $\tilde{\mathcal{X}} := \{xT : x \in \mathcal{X}\}$ be the sampled data domain. We assume the sampled encoder and decoder satisfy the analogues of Assumptions C.4–C.5 on $\tilde{\mathcal{X}}$ with constants $\tilde{L}_{enc}$ and $\tilde{L}_{dec}$. Due to the bi-Lipschitz property (Lemma D.6), these constants are related to the original ones by $\tilde{L}_{enc} \approx L_{enc}$ and $\tilde{L}_{dec} \approx L_{dec}$. Furthermore, we assume the sampled objective $\tilde{\mathcal{L}}$ satisfies the $(\tilde{\tau}_1, \tilde{\tau}_2)$-non-critical point condition in the region traversed by FedAvg iterates, consistent with the local geometry analysis in Appendix D.2.1.*

**Step 3: Convergence Guarantee.** We now complete the proof of Part A by invoking the general VAE properties.

*Proof of Part A.* The sampled objective function $\tilde{\mathcal{L}}(\tilde{W})$ has the identical functional form as the general VAE objective analyzed in Appendix D.2. By Theorem D.3 applied to $\tilde{\mathcal{L}}$ under Assumption D.7, $\tilde{\mathcal{L}}$ satisfies:

1. $(\tilde{a}, \tilde{b})$-semi-smoothness with $\tilde{a} = O(\tilde{L}_{\text{enc}})$ and $\tilde{b} = O(\tilde{L}_{\text{enc}}^2)$;

2. $(\tilde{\alpha}, \tilde{\beta})$-semi-Lipschitz continuity;

3. $(\tilde{\tau}_1, \tilde{\tau}_2)$-non-critical point condition.

Therefore, the sampled optimization problem satisfies all assumptions of the FedAvg convergence theorem (Theorem D.1). Applying Theorem D.1 to $\tilde{\mathcal{L}}$ yields:

$$\mathbb{E}[\tilde{\mathcal{L}}(\tilde{U}^{(R)})] - \tilde{\mathcal{L}}(\tilde{U}^*) \leq (1 - \lambda_1)^R \Delta_0 + 2\lambda_2, \tag{53}$$

where $\lambda_1, \lambda_2$ are defined as in Theorem D.1 using the sampled constants. Under Assumption D.7, this yields the standard FedAvg linear convergence guarantee for the sampled problem. $\qquad\square$

### D.4.2 Proof of Theorem 5.1, Part B

*Proof of Theorem 5.1, Part B.* We prove the total convergence guarantee by decomposing the error into optimization and sampling components. Let $U^* = \arg\min_U \mathcal{L}(U)$ denote the optimal parameters in the original $d$-dimensional space, and $\tilde{U}^* = \arg\min_{\tilde{U}} \tilde{\mathcal{L}}(\tilde{U})$ denote the optimal parameters in the sampled $s$-dimensional space.

By decomposing the total error using the triangle inequality, we have

$$E[\tilde{\mathcal{L}}(\tilde{U}^{(R)})] - \mathcal{L}(U^*) = \underbrace{E[\tilde{\mathcal{L}}(\tilde{U}^{(R)})] - \tilde{\mathcal{L}}(\tilde{U}^*)}_{\text{optimization error}} + \underbrace{\tilde{\mathcal{L}}(\tilde{U}^*) - \mathcal{L}(U^*)}_{\text{sampling approximation error}}$$

$$\leq (1 - \lambda_1)^R \Delta_0 + 2\lambda_2 + C \cdot \epsilon \cdot \mathcal{L}(U^*)$$

where the equality follows from adding and subtracting $\tilde{\mathcal{L}}(\tilde{U}^*)$. The desired inequality follows immediately from the result in Part A as well as Lemma D.5. This completes the proof.

$\qquad\square$

## E  Dataset

This study utilizes one simulated and three real-world scATAC-seq datasets to comprehensively evaluate our proposed framework. These datasets represent a range of complexities, dimensionalities, and biological contexts.

**Simulated scATAC-seq Dataset**  The simulated scATAC-seq datasets were generated using SCAN-ATAC-sim (Chen et al., 2021b), which efficiently down-samples bulk ATAC-seq profiles from ENCODE cell lines to create single-cell data with known ground truth labels. The simulation incorporates key parameters that reflect real experimental variations: signal-to-noise ratio (SNR), controlling the percentage of reads in true peak regions (ranging from 0.4 to 0.8), and sequencing depth, determining the average number of fragments per cell (ranging from 3000 to 8000). We simulated data for five distinct cell types from the hematopoietic system: Common Myeloid Progenitors (CMP), and their differentiated progeny Monocytes (MONO), Neutrophils (NEU), Megakaryocytes (MEGA), and Erythroid lineage cells (ERY), using bulk profiles from ENCODE. The resulting binary peak-by-cell matrix exhibits 92% sparsity across 90,635 features, capturing the inherent sparsity and high dimensionality characteristic of real scATAC-seq data. This simulated dataset provides an ideal benchmark for method evaluation as it offers ground truth labels while maintaining the technical complexities observed in experimental scATAC-seq data.

**Brain Prefrontal Cortex (PFC) Dataset**  The Brain PFC dataset (Xu et al., 2022) provides a challenging real-world test case from complex tissue. It consists of 21,045 single-cell chromatin accessibility profiles from post-mortem human prefrontal cortex (PFC) tissue, aggregated from three different donors. After standard processing and Harmony-based batch correction, the dataset comprises an ultra-high-dimensional feature space of 127,219 genomic regions with approximately 95% sparsity. The dataset encompasses nine distinct cell types, including major neuronal and glial classes. Its key characteristic for our evaluation is that these cell populations form distinct and well-separated clusters. This clear separation makes it an ideal case for validating a method's fundamental ability to correctly identify discrete cell identities in a heterogeneous environment before tackling more complex scenarios.

**PsychENCODE Dataset** The PsychENCODE dataset (Emani et al., 2024) is a large-scale scATAC-seq study of human post-mortem prefrontal cortex tissue, containing 96,673 cells profiled across an ultra-high-dimensional feature space of 423,443 genomic regions, with sparsity around 99%. The dataset captures the full cellular diversity of the brain, including major neuronal, glial, vascular, and immune cells. The chromatin accessibility patterns show characteristic scATAC-seq sparsity, with a right-skewed distribution of accessible regions per cell. With its massive feature dimension and complex cellular heterogeneity, this brain cell atlas serves as a critical benchmark for testing the scalability and robustness of computational methods in extreme-dimensional settings.

**10x Genomics PBMC Dataset** The PBMC (Peripheral Blood Mononuclear Cells) dataset is a comprehensive single-cell chromatin accessibility atlas from 10x Genomics, containing 11,493 cells profiled across 108,344 genomic regions, with sparsity around 93%. It captures 18 distinct immune cell populations, including numerous subtypes of T cells, B cells, and monocytes. Unlike the brain datasets, PBMC populations represent a continuous developmental hierarchy with transitional states between cell types. This results in less distinct cluster boundaries but biologically meaningful proximity between related cell populations, making it an ideal test for methods that must preserve both local and global structure in the learned representations.

# F Additional Experiment Details and Configurations

This appendix provides supplementary details for key experiments presented in the main text, focusing on the specific configurations and parameters that were omitted for brevity.

## F.1 Details for Confounded Heterogeneity Experiment

This experiment simulates a challenging federated scenario where each client's technical noise (SNR) is systematically correlated with its biological signal (cell-type distribution), creating a deliberately confounded, non-IID data partition. The configuration across $N = 5$ clients is detailed in Table 5, where each client has a unique SNR and a unique, skewed distribution over $K = 5$ cell types (MONO, NEU, CMP, MEGA, ERY). It directly tests the model's core capability to disentangle biological identity from client-specific technical artifacts.

Table 5: **Experimental setup for confounded heterogeneity analysis in federated learning.** Each client has a unique signal-to-noise ratio (SNR) and distinct cell-type distribution, creating correlated technical-biological heterogeneity.

| Client | SNR | Cell Type Distribution (cells) | | | | |
| --- | --- | --- | --- | --- | --- | --- |
| | | MONO | NEU | CMP | MEGA | ERY |
| Client 1 | 0.4 | 5,000 | 4,000 | 3,000 | 2,000 | 1,000 |
| Client 2 | 0.5 | 4,000 | 3,000 | 2,000 | 1,000 | 5,000 |
| Client 3 | 0.6 | 3,000 | 2,000 | 1,000 | 5,000 | 4,000 |
| Client 4 | 0.7 | 2,000 | 1,000 | 5,000 | 4,000 | 3,000 |
| Client 5 | 0.8 | 1,000 | 5,000 | 4,000 | 3,000 | 2,000 |

## F.2 Details for Extreme Class Imbalance Experiment

To evaluate FL-Sailer's resilience against the "minority class collapse" problem common in distributed learning, we designed a synthetic dataset exhibiting severe class imbalance, mirroring real-world hematopoietic differentiation hierarchies. In this setting, abundant differentiated cells often dominate the gradient updates, potentially erasing the signal of rare progenitor populations.

**Dataset Configuration.** We constructed a dataset comprising 23,500 cells with a **16-fold imbalance** between the majority and minority classes. The population distribution is detailed in Table 6. The Common

Myeloid Progenitor (CMP) population (500 cells, $\approx 2.1\%$) serves as the critical "stress test." Biologically, CMPs are the upstream progenitors that give rise to the four differentiated lineages (MONO, NEU, MEGA, ERY). Consequently, the model's ability to distinguish CMPs from their progeny is not merely a classification task but a test of its ability to preserve developmental structure amidst statistical noise.

**Experimental Protocol.** We evaluated this imbalanced configuration across three distinct Signal-to-Noise Ratios (SNR $\in \{0.4, 0.6, 0.8\}$) to decouple the effects of sparsity from class imbalance. The data was partitioned across $N = 5$ clients. This setup rigorously tests whether the global model aggregation in FL-Sailer can preserve rare signals that might be locally insignificant or statistically drowned out by majority classes during local training.

Table 6: **Cell population statistics for the extreme class imbalance experiment.** The dataset simulates a realistic bone marrow environment where differentiated cells vastly outnumber progenitors.

| Cell Type | Biological Role | Count | Proportion | Imbalance Ratio (vs CMP) |
|---|---|---|---|---|
| MONO | Monocytes (Differentiated) | 8,000 | 34.0% | 16.0$\times$ |
| NEU | Neutrophils (Differentiated) | 5,000 | 21.3% | 10.0$\times$ |
| MEGA | Megakaryocytes (Differentiated) | 5,000 | 21.3% | 10.0$\times$ |
| ERY | Erythroid (Differentiated) | 5,000 | 21.3% | 10.0$\times$ |
| CMP | Progenitor (Rare) | 500 | 2.1% | 1.0$\times$ |

### F.3 Visualization of sequencing depth in FL-Sailer embeddings

To illustrate how FL-Sailer handles sequencing-depth variation, Figure 4 visualizes UMAP embeddings from FL-Sailer colored by raw per-cell fragment counts for both the heterogeneous-depth synthetic experiment in Table 1 (left) and the three real scATAC-seq datasets (right) corresponding to the experiment in Figure 3. Within each annotated cell-type cluster, cells span the full range of sequencing depths and are broadly intermingled, and we do not observe clear depth-driven subclusters or gradients, indicating that the learned embeddings are largely invariant to sequencing depth.

### F.4 Invariant VAE

Table 3 and Figure 3 discussed the effect of adaptive sampling. In this subsection, we study the contribution of the invariance term. As shown in Table 7, adding invariance consistently improves clustering quality in both centralized and federated settings, with substantially larger gains in the federated model. Although the invariant objective is inherited from SAILER (Cao et al., 2021) rather than newly introduced in this work, these results show that it remains a crucial component in the federated setting, where suppressing client-specific technical variation is necessary to fully realize the benefits of adaptive sampling.

### F.5 Wall-clock runtime analysis.

To complement the communication analysis in Section 6.2, we report a detailed wall-clock runtime breakdown for FL-Sailer in Table 8. We used Nvidia RTX A6000 48GB for Brain PFC and PsychENCODE dataset centralized and federated training, and Nvidia RTX 3090 24GB for PBMC dataset centralized and federated training. The key observations are:

**Measurement protocol.** For FL-Sailer, we log (i) the sum over rounds of the slowest client's local training time ("client compute, critical path"), (ii) the total federated training time after all clients have finished data loading ("FL training time"), and (iii) the non-training portion of the federated phase, which captures communication, server-side aggregation, and cross-client synchronization overheads. For centralized SAILER, we report total training time on a single GPU, excluding initial data loading.

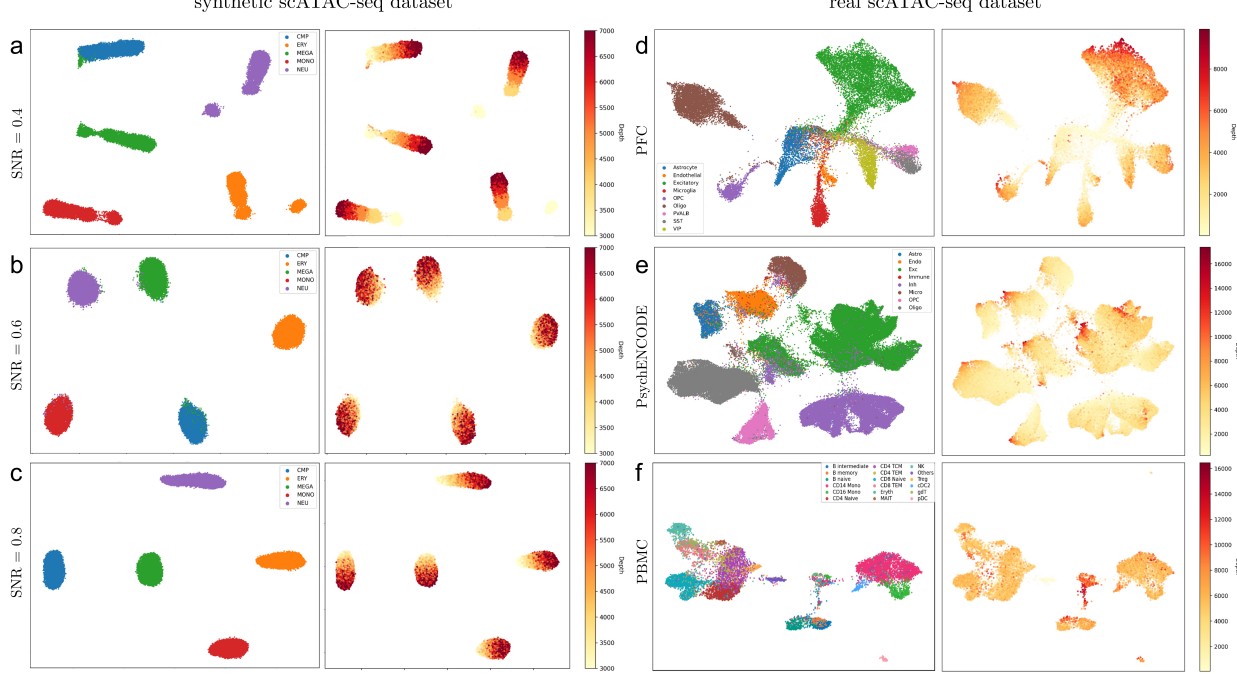

Figure 4: Sequence depth analysis of FL-Sailer clustering performance on synthetic (left) and real-world (right) scATAC-seq datasets.
Left panels: (a) SNR = 0.4 (ARI = 0.858, Silhouette = 0.740), (b) SNR = 0.6 (ARI = 0.981, Silhouette = 0.846), and (c) SNR = 0.8 (ARI = 1.000, Silhouette = 0.849).
Right panels: (d) PFC (ARI = 0.800, Silhouette = 0.453), (e) PsychENCODE (ARI = 0.778, Silhouette = 0.363), and (f) PBMC (ARI = 0.742, Silhouette = 0.227).

Table 7: **Ablation of invariant VAE.**

| Dataset | Model variant | ARI | Silhouette |
|---|---|---|---|
| Brain PFC | Centralized SAILER (with sampling, no invariance) | $0.689 \pm 0.010$ | $0.436 \pm 0.004$ |
| | Centralized SAILER (with sampling, with invariance) | $0.702 \pm 0.008$ | $0.466 \pm 0.002$ |
| | FL-Sailer (with sampling, no invariance) | $0.634 \pm 0.005$ | $0.434 \pm 0.003$ |
| | FL-Sailer (with sampling, with invariance) | $0.855 \pm 0.005$ | $0.463 \pm 0.002$ |
| PsychENCODE | Centralized SAILER (with sampling, no invariance) | $0.605 \pm 0.002$ | $0.411 \pm 0.002$ |
| | Centralized SAILER (with sampling, with invariance) | $0.624 \pm 0.002$ | $0.423 \pm 0.002$ |
| | FL-Sailer (with sampling, no invariance) | $0.629 \pm 0.001$ | $0.301 \pm 0.002$ |
| | FL-Sailer (with sampling, with invariance) | $0.781 \pm 0.001$ | $0.378 \pm 0.001$ |
| PBMC | Centralized SAILER (with sampling, no invariance) | $0.601 \pm 0.007$ | $0.107 \pm 0.005$ |
| | Centralized SAILER (with sampling, with invariance) | $0.674 \pm 0.006$ | $0.156 \pm 0.003$ |
| | FL-Sailer (with sampling, no invariance) | $0.620 \pm 0.007$ | $0.144 \pm 0.005$ |
| | FL-Sailer (with sampling, with invariance) | $0.742 \pm 0.005$ | $0.230 \pm 0.003$ |

**Client computation vs. communication/overhead.** At the sampling rates used in our main experiments ($\rho = 0.15$ for Brain PFC, $\rho = 0.20$ for PsychENCODE, and $\rho = 0.30$ for PBMC), critical-path client computation accounts for only about 20–50% of the federated training time:

- Brain PFC ($\rho = 0.15$): 20.1 min compute vs. 41.8 min total;

- PsychENCODE ($\rho = 0.20$): 1.8 h compute vs. 8.8 h total;

- PBMC ($\rho = 0.30$): 16.4 min compute vs. 48.7 min total.

The remaining 50–80% of the wall-clock time is attributable to communication and system overheads, quantitatively confirming that communication and cross-client orchestration, rather than raw client compute, are the main bottlenecks in federated scATAC-seq.

**Effect of sampling rate $\rho$.** Varying $\rho$ exhibits the expected trade-off between runtime and accuracy. Increasing $\rho$ generally increases both non-training time and total training time, since more parameters must be transmitted and aggregated each round, while providing diminishing or even negative returns in clustering performance beyond $\rho \approx 0.2$–$0.3$ (see Tables 4 and 8). At the optimal values ($\rho = 0.15/0.20/0.30$), FL-Sailer simultaneously achieves the best ARI, an approximate 80% reduction in communication, and substantial reductions in end-to-end wall-clock time relative to centralized SAILER.

Table 8: **Wall-clock runtime of FL-Sailer across sampling rates $\rho$ on three real scATAC-seq datasets.** We report mean and max client training time, mean and max data-loading time, total federated training time (excluding initial data loading), and non-training time (communication and system overhead).

| Dataset | $\rho$ | Client training (min) | | Data loading (min) | | Total training | Non-training |
|---|---|---|---|---|---|---|---|
| | | mean | max | mean | max | (min, excl. load) | (min) |
| Brain PFC | 0.10 | 10.75 | 23.56 | 0.85 | 0.87 | 36.28 | 12.72 |
| | 0.15 | 9.79 | 20.14 | 0.90 | 1.05 | 41.80 | 21.66 |
| | 0.20 | 10.08 | 22.51 | 0.82 | 0.92 | 43.42 | 20.91 |
| | 0.25 | 9.32 | 21.20 | 0.83 | 0.83 | 47.87 | 26.67 |
| | 0.30 | 9.57 | 20.76 | 0.90 | 0.92 | 52.32 | 31.55 |
| | 0.35 | 9.68 | 20.84 | 0.98 | 0.98 | 63.00 | 42.16 |
| | 0.40 | 9.24 | 20.23 | 0.85 | 0.87 | 70.07 | 49.84 |
| | 0.45 | 9.82 | 21.54 | 1.06 | 1.07 | 72.40 | 50.86 |
| | 0.50 | 9.74 | 22.32 | 1.08 | 1.08 | 79.43 | 57.11 |
| | 1.00 | 12.69 | 28.74 | 0.82 | 0.83 | 89.20 | 60.46 |
| PsychENCODE | 0.10 | 42.51 | 95.00 | 1.06 | 1.13 | 327.88 | 232.88 |
| | 0.15 | 42.59 | 97.45 | 4.10 | 4.75 | 424.97 | 327.51 |
| | 0.20 | 45.84 | 108.45 | 2.35 | 2.73 | 527.78 | 419.33 |
| | 0.25 | 56.15 | 127.89 | 4.92 | 5.77 | 620.25 | 492.36 |
| | 0.30 | 51.97 | 119.35 | 4.53 | 5.23 | 648.83 | 529.49 |
| | 0.35 | 62.81 | 144.92 | 2.02 | 2.27 | 730.10 | 585.17 |
| | 0.40 | 66.26 | 153.27 | 4.51 | 5.22 | 739.72 | 586.45 |
| | 0.45 | 73.76 | 171.84 | 7.75 | 8.97 | 835.80 | 663.96 |
| | 0.50 | 73.92 | 171.78 | 0.85 | 0.87 | 826.17 | 654.39 |
| | 1.00 | 50.29 | 110.28 | 1.72 | 1.73 | 774.82 | 664.54 |
| PBMC | 0.10 | 10.35 | 19.99 | 0.96 | 1.03 | 34.97 | 14.98 |
| | 0.15 | 11.28 | 20.92 | 0.98 | 1.08 | 39.20 | 18.28 |
| | 0.20 | 9.17 | 17.16 | 1.06 | 1.17 | 42.47 | 25.31 |
| | 0.25 | 9.10 | 17.78 | 1.11 | 1.23 | 44.77 | 26.98 |
| | 0.30 | 8.95 | 16.43 | 1.24 | 1.45 | 48.72 | 32.29 |
| | 0.35 | 8.79 | 16.80 | 1.17 | 1.35 | 51.38 | 34.58 |
| | 0.40 | 8.68 | 16.88 | 1.31 | 1.47 | 54.10 | 37.22 |
| | 0.45 | 8.67 | 16.69 | 1.07 | 1.18 | 55.28 | 38.59 |
| | 0.50 | 8.82 | 15.61 | 1.23 | 1.43 | 56.70 | 41.09 |
| | 1.00 | 10.13 | 19.75 | 0.78 | 0.83 | 71.18 | 51.44 |

Centralized SAILER training: Brain PFC: 609 min, PsychENCODE: 3252 min, PBMC: 372 min.

