# OpenReview forum: "FL-Sailer: Efficient and Privacy-Preserving Federated Learning for Scalable Single-Cell Epigenetic Data Analysis via Adaptive Sampling"
_TMLR — Accepted by TMLR_

### Review · Reviewer_VzVx · 2026-01-30

**Summary Of Contributions:**

The paper introduces a federated approach based on Sailer, an approach for single-cell epigenomic sequencing. The federated approach helps with privacy by processing data on the clients and only sending a lower-dimensional feature vector to the global model. This also results in a smaller communication cost, and the federated setting is shown to help against confounding, surpassing the centralized approach on multiple confounded benchmarks.

**Additional Comments:**

It would be great if the authors could provide their code.

The figures would benefit from larger text, especially Figure 1.

**Audience:**

Yes

**Audience Explanation:**

I have no background in single-cell analysis or similar fields, so I find it difficult to assess the potential interest for this paper. On the technical side, the use of federated learning and VAEs to combine different sources of information while accounting for heterogeneity due to confounding is certainly interesting.

**Broader Impact Concerns:**

I do not have any ethical concerns regarding this paper.

**Claims And Evidence:**

Yes

**Claims Explanation:**

The methodology has a solid theoretical basis with convergence guarantees.

Experiments show good results for FL-Sailer on the following points

- FL-Sailer performs well (better than Sailer) on synthetic problems with varying signal-to-noise ratio (Figure 2) and heterogeneous sequencing depth (Table 1)
- FL-Sailer helps overcome confounding due to heterogeneity and can handle class imbalance (Figure 2)
- FL-Sailer successfully reduces the communication cost from clients to the server (Table 2)
- FL-Sailer outperforms other federated methods and the centralized approach on real-world datasets (Table 4, Figure 3)

It is also shown how sampling is beneficial both in terms of communication cost and  model performance (Table 3).
Additionally, the clustering visualizations of Figures 2 and 3 help illustrate the results and clusters produced by FL-Sailer in an intuitive manner.

**Requested Changes:**

1. There are a few point that would benefit from a brief introduction / clarification. Could you please explain the ARI and silhouette scores in more detail? Can you elaborate on the LSI baseline used in Figure 2? How does the computation time of federated learning and centralized learning differ (communication has been analyzed, but what about the computation on each client)?
2. The analysis and experiments focus on the efficiency and performance of the model. My concern is regarding privacy. How does FL-Sailer ensure that no sensitive information about the original data can be recovered from the communicated features?

---

> ### Author Response · Authors · 2026-02-05
> **Response Part 1**
>
> **Clarification of ARI and silhouette scores.**
> We thank the reviewer for noting that the definitions of the clustering metrics were too terse.
>
> The *Adjusted Rand Index (ARI)* measures how well the clustering obtained from the learned embeddings recovers the known cell-type labels. It counts, over all pairs of cells, how many are assigned consistently (both in the same cluster or both in different clusters) by the clustering and by the ground-truth labels, and subtracts the expected agreement of a random clustering. Thus $\mathrm{ARI} = 0$ corresponds to chance level, $\mathrm{ARI} = 1$ to perfect recovery, and negative values indicate worse-than-random partitions.
>
> The *silhouette score* quantifies cluster compactness and separation without using labels. For each cell $i$, we compute $a(i)$, the average distance to other cells in the same cluster, and $b(i)$, the smallest average distance to cells in any other cluster. The silhouette of cell $i$ is
> $$
> s(i) = \frac{b(i) - a(i)}{\max\{a(i), b(i)\}},
> $$
> which lies in $[-1, 1]$. Values close to $1$ indicate that the cell is closer to its own cluster than to others, values near $0$ correspond to overlapping clusters, and negative values indicate mis-clustering. We report the dataset-level silhouette as the average of $s(i)$ over all cells.
>
> ---
>
> **Details of the LSI baseline in Figure 2.**
> The ``LSI'' baseline in Figure~2 follows a standard scATAC-seq preprocessing pipeline used in tools such as SnapATAC and Signac. We start from the binary peak-by-cell matrix, apply TF--IDF normalization, and then perform truncated SVD:
>
> * For peak $p$ and cell $j$, we compute
>
> $$
> \mathrm{TF}_{pj} = \frac{x_{pj}}{\sum_{q} x_{qj}}
> $$
>
> $$
> \mathrm{IDF}_p = \log\frac{1 + n}{1 + n_p}
> $$
> where $n$ is the number of cells and $n_p$ is the number of cells in which peak $p$ is accessible.
> * We construct the TF--IDF matrix and compute a truncated SVD, keeping the top $K$ left singular vectors as the LSI embedding.
> * We then run clustering and evaluate ARI and silhouette scores on the resulting clusters.
>
> This baseline corresponds to a widely used linear dimensionality-reduction and clustering pipeline for scATAC-seq data, and serves as a strong, interpretable baseline for comparing FL-Sailer on synthetic data.
>
> ---
>
> **Privacy considerations and potential leakage from communicated quantities.**
> We appreciate this important question. Our goal in this work is to address the practical privacy and governance constraints that prevent institutions from sharing raw scATAC-seq profiles. We adopt a standard cross-silo federated learning threat model: raw cell-by-peak matrices never leave the local institution; the coordinating server is honest-but-curious; and communication channels are protected using transport-layer encryption. Under this model, FL-Sailer reduces privacy risk in three ways:
>
> 1. **No per-cell profiles are transmitted.** In Phase 1, each client sends only approximate column leverage scores $\hat{\ell}^{(i)} \in \mathbb{R}^d$ and its sample size $n_i$. These scores are randomized summaries of the local matrix $A_i$ obtained via a Gaussian sketch and thin QR factorization. Recovering individual binary accessibility profiles from $\hat{\ell}^{(i)}$ is an underdetermined inverse problem: the server observes only $d$ aggregated statistics instead of the full $n_i \times d$ matrix.
> 2. **Model updates are aggregated, not raw features.** In Phase 2, clients communicate parameter deltas $\Delta W_i$, that is, gradients averaged over mini-batches of many cells in the sampled feature space. These updates have similar information content to standard cross-silo FL systems and never expose raw peak-level vectors. In practice, each client holds thousands of cells, so each gradient update is a heavily averaged statistic rather than a single-cell signal.
> 3. **Dimensionality reduction further limits reconstruction.** The global server only observes updates in the sampled subspace of size $s = \rho d$. The combination of leverage-score sampling and invariant representation learning acts as an information bottleneck: we explicitly minimize the mutual information $I(z; c)$ between latent representations and technical covariates, and empirically observe that the learned embeddings discard much of the idiosyncratic technical variation present in raw profiles.
>
> Like most cross‑silo FL systems, FL‑Sailer does not by itself provide a formal DP or cryptographic guarantee. Its privacy benefits from: 1)raw cell‑by‑peak matrices and per‑cell profiles stay local, and 2) the server only receives aggregate leverage‑score statistics and averaged gradients in a reduced feature space. Stronger mechanisms (e.g., client‑side DP or secure aggregation) are fully compatible with our framework can be added in the future.

---

> ### Author Response · Authors · 2026-02-05
> **Response Part 2**
>
> **Dataset scale, communication cost, and computation time of FL-Sailer vs. centralized SAILER.**
>
> We thank the reviewer for raising questions about the practical runtime of FL-Sailer and for noting that the scale of the biological datasets and evaluation metrics may not be fully clear from the current draft. In this response, we (i) clarify the dimensionality of our real scATAC-seq datasets, (ii) explain how this dimensionality translates into communication cost in our chromosome-block VAE architecture, and (iii) provide a detailed wall-clock comparison between federated and centralized training.
>
> **Dataset scale and intuition.**
> Our three real scATAC-seq datasets are extremely high-dimensional:
>
> - Brain PFC: 21,045 cells $\times$ 127,219 peaks ($\approx$ 95% zeros);
> - PsychENCODE: 96,673 cells $\times$ 423,443 peaks ($\approx$ 99% zeros);
> - PBMC: 11,493 cells $\times$ 108,344 peaks ($\approx$ 93% zeros).
>
> For example, the conceptual cell-by-peak matrix for PsychENCODE has about $9.7\times 10^4 \times 4.2\times 10^5 \approx 4\times 10^{10}$ entries. Even if we stored a single dense 8-bit copy, this would already be on the order of tens of GB. Transferring such matrices between institutions is therefore not realistic, which is why our method focuses on communicating model parameters rather than raw data.
>
> **Why large datasets still lead to large communication in FL.**
>
> In FL-Sailer, as in SAILER (Cao et al., 2021), we use a chromosome-block VAE: peaks from each chromosome (22 autosomes $+$ X) are fed into a separate encoder--decoder block. This biologically motivated architecture implies that the number of model parameters grows approximately linearly with the number of peaks that are kept after sampling. With sampling rate $\rho$ and total number of peaks $d$, the total parameter count $|W|$ is therefore roughly proportional to $\rho d$. Since standard federated optimization transmits a full model (or update) in each round, the per-round communication scales as $O(|W|)$, and in our setting $|W| \approx \Theta(\rho d)$.
>
> This scaling is reflected in Table 2 of the paper: for PsychENCODE, the no-sampling baseline ($\rho = 1.0$) uses 54.75M parameters, corresponding to 208.9 MB per round and 20.4~GB over 100 rounds; setting $\rho = 0.2$ reduces the model to 11.06M parameters, 42.2 MB per round, and 4.1 GB total communication (about $80$ % reduction). Similar $\sim$ 80% reductions hold for Brain PFC and PBMC. Thus, although FL transmits weights rather than data, in extreme-dimensional scATAC-seq the number of weights itself is very large, and naive FL without sampling would incur prohibitive model-update communication.
>
>
> **Runtime measurement protocol.**
> Our initial submission focused on communication-theoretic analysis and clustering performance (Tables 2--4) and therefore did not report explicit wall-clock training times.
> In response to the reviewer’s suggestion, we have now instrumented both centralized SAILER and FL-Sailer to measure runtime on the same GPU hardware.
> These wall-clock times reflect a realistic cross-silo setup with one GPU per client; centralized SAILER uses a single GPU. For Brain PFC and PyschENCODE dataset we have 5 federated clients whereas PBMC dataset we have 3 federated clients. The numbers therefore capture both algorithmic efficiency from adaptive sampling and practical gains from distributed computation.
>
> For FL-Sailer, we record:
> 1. the sum over communication rounds of the slowest client’s local training time, $\sum_r \max_j T_j(r)$ (``client compute, critical path'');
> 2. the total federated training time after all clients have finished data loading (``FL training time'');
> 3. the non-training time, it means the portion of the federated training phase that is not spent on client-side model updates. This quantity captures time consumed by communication (broadcasting and uploading model parameters), server-side aggregation, and cross-client synchronization and scheduling overheads.
>
> For centralized SAILER, we report the total training time on a single GPU, excluding initial data loading, which is dominated by model computation since there is no cross-client communication.

---

> ### Author Response · Authors · 2026-02-05
> **Response Part 3**
>
> **Client computation vs. communication/overhead.**
> Decomposing the FL runtime shows that client-side computation is not the dominant cost. At the optimal $\rho$, critical-path client compute accounts for only about 20--50% of the total federated training time, with the remaining 50-80% attributable to communication and system overhead:
>
> - Brain PFC ($\rho = 0.15$): 20.1 min compute vs. 41.8 min total;
> - PsychENCODE ($\rho = 0.20$): 1.8 h compute vs. 8.8 h total;
> - PBMC ($\rho = 0.30$): 16.4 min compute vs. 48.7 min total.
>
> This quantitatively confirms our main design intuition: in federated scATAC-seq, communication and cross-client orchestration are the main bottlenecks, and our adaptive leverage-score sampling (together with the chromosome-block architecture) is crucial for making FL both feasible and efficient at this scale.
>
> **Effect of sampling rate $\rho$.**
> Finally, we note that varying $\rho$ exhibits the expected trade-off between runtime and accuracy.
> As expected, increasing $\rho$ generally increases both non-training time and total training time, since more parameters must be transmitted and aggregated each round, while providing diminishing or even negative returns in clustering performance beyond $\rho \approx 0.2$--$0.3$ (see Table 3).
> At the optimal $\rho$ values (0.15 for Brain PFC, 0.20 for PsychENCODE, 0.30 for PBMC), FL-Sailer simultaneously achieves the best ARI, $\approx$ 80% reduction in communication, and substantial reductions in end-to-end training time relative to centralized SAILER.
>
> We will add a concise table with these runtime numbers to the revised manuscript and clarify the dataset scale and parameter--communication relationship in Section~6.2 of the paper.
>
> **Code availability and figure readability.**
> We implemented FL-Sailer by extending the publicly available SAILER codebase (Cao et al., 2021) and will release our implementation upon acceptance to facilitate reproducibility.
>
> **Figure readability.** Thank you for pointing this out. In the revised manuscript we will increase the font sizes and line widths in Figures 1--3, and verify that all text is legible when printed.
>
>
>
> **Running time results**
>
> **Brain PFC**
>
> | ρ    | Client training (mean, min) | Client training (max, min) | Data loading (mean, min) | Data loading (max, min) | Total training time (min, excl. load) | Non-training time (min) |
> |------|-----------------------------|----------------------------|--------------------------|-------------------------|----------------------------------------|-------------------------|
> | 0.10 | 10.75                       | 23.56                      | 0.85                     | 0.87                    | 36.28                                  | 12.72                   |
> | 0.15 | 9.79                        | 20.14                      | 0.90                     | 1.05                    | 41.80                                  | 21.66                   |
> | 0.20 | 10.08                       | 22.51                      | 0.82                     | 0.92                    | 43.42                                  | 20.91                   |
> | 0.25 | 9.32                        | 21.20                      | 0.83                     | 0.83                    | 47.87                                  | 26.67                   |
> | 0.30 | 9.57                        | 20.76                      | 0.90                     | 0.92                    | 52.32                                  | 31.55                   |
> | 0.35 | 9.68                        | 20.84                      | 0.98                     | 0.98                    | 63.00                                  | 42.16                   |
> | 0.40 | 9.24                        | 20.23                      | 0.85                     | 0.87                    | 70.07                                  | 49.84                   |
> | 0.45 | 9.82                        | 21.54                      | 1.06                     | 1.07                    | 72.40                                  | 50.86                   |
> | 0.50 | 9.74                        | 22.32                      | 1.08                     | 1.08                    | 79.43                                  | 57.11                   |
> | 1.0  | 12.69                       | 28.74                      | 0.82                     | 0.83                    | 89.20                                  | 60.46                   |
>
> Brain PFC centralized training time: 609 min.

---

> ### Author Response · Authors · 2026-02-05
> **Response Part 4**
>
> **PsychENCODE**
>
> | ρ    | Client training (mean, min) | Client training (max, min) | Data loading (mean, min) | Data loading (max, min) | Total training time (min, excl. load) | Non-training time (min) |
> |------|-----------------------------|----------------------------|--------------------------|-------------------------|----------------------------------------|-------------------------|
> | 0.10 | 42.51                       | 95.00                      | 1.06                     | 1.13                    | 327.88                                 | 232.88                  |
> | 0.15 | 42.59                       | 97.45                      | 4.10                     | 4.75                    | 424.97                                 | 327.51                  |
> | 0.20 | 45.84                       | 108.45                     | 2.35                     | 2.73                    | 527.78                                 | 419.33                  |
> | 0.25 | 56.15                       | 127.89                     | 4.92                     | 5.77                    | 620.25                                 | 492.36                  |
> | 0.30 | 51.97                       | 119.35                     | 4.53                     | 5.23                    | 648.83                                 | 529.49                  |
> | 0.35 | 62.81                       | 144.92                     | 2.02                     | 2.27                    | 730.10                                 | 585.17                  |
> | 0.40 | 66.26                       | 153.27                     | 4.51                     | 5.22                    | 739.72                                 | 586.45                  |
> | 0.45 | 73.76                       | 171.84                     | 7.75                     | 8.97                    | 835.80                                 | 663.96                  |
> | 0.50 | 73.92                       | 171.78                     | 0.85                     | 0.87                    | 826.17                                 | 654.39                  |
> | 1.0  | 50.29                       | 110.28                     | 1.72                     | 1.73                    | 774.82                                 | 664.54                  |
>
> PsychENCODE centralized training time: 3252 min.
>
> ---
>
> **PBMC**
>
> | ρ    | Client training (mean, min) | Client training (max, min) | Data loading (mean, min) | Data loading (max, min) | Total training time (min, excl. load) | Non-training time (min) |
> |------|-----------------------------|----------------------------|--------------------------|-------------------------|----------------------------------------|-------------------------|
> | 0.10 | 10.35                       | 19.99                      | 0.96                     | 1.03                    | 34.97                                  | 14.98                   |
> | 0.15 | 11.28                       | 20.92                      | 0.98                     | 1.08                    | 39.20                                  | 18.28                   |
> | 0.20 | 9.17                        | 17.16                      | 1.06                     | 1.17                    | 42.47                                  | 25.31                   |
> | 0.25 | 9.10                        | 17.78                      | 1.11                     | 1.23                    | 44.77                                  | 26.98                   |
> | 0.30 | 8.95                        | 16.43                      | 1.24                     | 1.45                    | 48.72                                  | 32.29                   |
> | 0.35 | 8.79                        | 16.80                      | 1.17                     | 1.35                    | 51.38                                  | 34.58                   |
> | 0.40 | 8.68                        | 16.88                      | 1.31                     | 1.47                    | 54.10                                  | 37.22                   |
> | 0.45 | 8.67                        | 16.69                      | 1.07                     | 1.18                    | 55.28                                  | 38.59                   |
> | 0.50 | 8.82                        | 15.61                      | 1.23                     | 1.43                    | 56.70                                  | 41.09                   |
> | 1.0  | 10.13                       | 19.75                      | 0.78                     | 0.83                    | 71.18                                  | 51.44                   |
>
> PBMC centralized training time: 372 min.

---

> > ### Author Response · Authors · 2026-02-23
> > **Response**
> >
> > Dear Reviewer VzVx, we think this might be a mistake, we didn't set the visibility correctly for our response. If you have any questions, please feel free to ask, thank you.

---

> > > ### Comment · Reviewer_VzVx · 2026-03-02
> > >
> > > I thank the authors for their response. The results on computation time match the theory and help showcase the efficiency of the federated approach, as well as the role of the sampling rate. I also appreciate the clarifications with respect to the metrics and the privacy considerations.
> > > I have only one remaining question regarding the privacy considerations response. You state that you "empirically observe that the learned embeddings discard much of the idiosyncratic technical variation present in raw profiles". Can you please elaborate on this point, i.e., what is the "technical variation" and what did you observe? For example, can you describe this observation using the plots shown in Figure 2 or 3?

---

> > > > ### Author Response · Authors · 2026-03-12
> > > > **Response**
> > > >
> > > > We thank the reviewer for this follow-up question.
> > > >
> > > > By "technical variation" we mean non-biological covariates such as per-cell sequencing
> > > > depth / fragment counts and donor / batch / institution (which coincide with clients in our FL setup). These are modeled as technical confounders $c$ in the conditional decoder
> > > > $p_\phi(x \mid z, c)$. In the synthetic experiments we additionally vary a client-specific SNR parameter to simulate extra technical noise.
> > > >
> > > > Empirically, we observe that these covariates no longer organize the FL-Sailer latent space. In the confounded-heterogeneity experiment in Figure 2(b), each client has a distinct SNR and a skewed cell-type distribution. In the FL-Sailer panel, when we color the UMAP by "cell type", each population forms a compact cluster, and when we color by "dataset/client" the points from all five clients are well mixed within each cell-type cluster rather than forming client-specific clusters. This indicates that the dominant axes of variation in the learned embedding align with cell identity rather than with the client/SNR technical covariate.
> > > >
> > > > In the heterogeneous-depth simulation underlying Table 1, cells within each cell-type cluster span roughly the 3000–8000 fragment range, with different depths broadly mixed inside each cluster. On the three real datasets in Figure 3, the FL-Sailer embeddings similarly contain cells across the observed depth ranges (e.g., approximately 1000–9000 for Brain PFC and 1000–17000 for PsychENCODE and PBMC) within each cell-type cluster, and we do not observe clear depth-driven subclusters. See the newly added depth-colored FL-Sailer visualizations in Appendix F.3.

---

> > > > > ### Comment · Reviewer_VzVx · 2026-03-16
> > > > >
> > > > > I thank the authors for their additional explanations. I have no further remaining concerns or questions.

---

### Review · Reviewer_ScDv · 2026-02-23

**Summary Of Contributions:**

The authors propose **FL-Sailer**, a federated learning (FL) framework for single-cell ATAC-seq (scATAC-seq) analysis. The manuscript is motivated by how privacy concerns and data size prevent multi-institutional data sharing, while standard FL methods (or even methods for scRNA-seq) are ill-suited for the sparsity, high dimensionality, and institution-specific data heterogeneity. The main contribution of this work is a federated extension to the Sailer method that are designed to meet the challenges outlined by the authors (namely, privacy-preserving, sparsity, high dimensionality, and heterogeneity). The authors conduct experiments with the standard Sailer method, supply proofs for convergence guarantees, and demonstrate robustness to a variety of sources of heterogeneity. My main concerns are around narrative and justification of their motivation (such as the lack of a principled argument for the inappropriateness of PCA, lack of details on privacy claims etc.). All in all, the paper is well motivated, clearly written, and the method and results are of interest to the TMLR audience.

**Audience:**

Yes

**Audience Explanation:**

The general thrust of the motivation of the work (sparsity, heterogeneity, and dimenstionality paired with the need for privacy) are solid, sharpening the narrative and positioning could help readers (see below).

**Claims And Evidence:**

Yes

**Claims Explanation:**

While the theoretical justification seems solid and the empirical results demonstrate performance uplift compared to other methods, the authors should include code, details on training/inference hardware, and some measures of variation for performance metrics (and associated statistical significance tests on comparing performance).

**Requested Changes:**

Major:
- **More details on motivation for biologically informative genomic regions.** In the introduction, the authors write “[u]nlike PCA which creates uninterpretable linear combinations, our sampling module directly selects biologically informative genomic regions,” and demonstrate this informative region selection empirically in their experiments in Section 6 and  theoretically in Appendix B. However, despite surfacing PCA early, it doesn’t seem like the manuscript empirically demonstrates how PCA fails (e.g. via silhouette scores). Adding this context can contextualize how simple methods fail to capture these informative regions may strengthen the narrative.
- **Justification of inappropriateness of scRNA-seq FL methods.** In section 2.2 the authors point to scPrivacy and scFed as extant methods for transcriptomic analysis and characterize them as “primarily tailored for lower-dimensional, less sparse modalities” inappropriate for the sparsity and dimensionality of scATAC-seq (i.e. the $10^5-10^7$ features). The authors should contextualize the dimensionality of scRNA-seq methods more directly (e.g. ~20k for the datasets used in scFed paper). More importantly, while the motivation of higher sparsity and dimensionality is valid, the authors should include a formal justification or empirical demonstration of how extant methods for scRNA-seq fails for scATAC-seq.
- **Deeper analysis on privacy claims** Given that privacy is one of the main motivators of the work, the authors should flesh out the details of the privacy claims they are making as well as some empirical results (e.g. resistance to membership inference).
- **Add interval estimates for performance metrics.** The performance metrics in Section 6 are all point estimates. While the authors include cluster plots in Figure 3, adding some notion of interval estimates (e.g. through bootstrapped confidence intervals for silhouette scores) in the tables will help further contextualize their results, particularly in Table 4 when comparing across different federated methods.
- **Add hardware details.** While the manuscript includes details on communication efficiency, the authors should add details on training/inference hardware and how long training took. Additionally, some details on the implications of the compute hardware for future users of the system would be welcome.
- **Code repository** The authors should consider adding an (anonymized) code repository.

Minor:
- Add a space between “Federated learning” and (FL) in the first line of the last paragraph on page 1.

---

> ### Author Response · Authors · 2026-03-12
> **Response**
>
> We thank Reviewer ScDv for the careful reading and helpful suggestions.
>
> (1) PCA / biologically informative regions.
> We thank the reviewer for this suggestion. We agree that our earlier wording was too strong. Our goal is not to present a separate empirical study of PCA failure. Rather, our point is that leverage-score sampling selects actual genomic regions, which provides direct biological interpretability, whereas dense linear projections do not. We have therefore revised the manuscript to remove the direct PCA comparison and to state this point more carefully.
>
> (2) scRNA-seq FL methods.
> We agree and have revised Section 2.2 accordingly. We now describe scPrivacy and scFed as important related works for scRNA-seq, while clarifying that our setting is ultra-high-dimensional and highly sparse scATAC-seq (108K--423K regions; 93\%--99\% sparsity), which induces a different communication and optimization regime. We adopt SAILER as the backbone because it is designed for scATAC-seq data.
>
>
> (3) Privacy claims.
> We have modified the corresponding wording in the manuscript Section 3.3. Our claim is limited to the standard cross-silo setting in which raw profiles are not centralized; we do not claim formal differential privacy or resistance to specific attack models in this work.
>
> (4) Variability.
> We now report mean $\pm$ standard deviation via bootstrap resampling over cell embeddings for all main tables. See updated manuscript.
>
> (5) Runtime.
> Thank you for pointing this out. We added runtime results and hardware specs in the Appendix F.5 to complement the communication analysis. You can also refer to the response to Reviewer VzVx.
>
> (6) Code.
> We plan to release the implementation upon publication.
>
> We also fixed the minor spacing issue in “Federated learning (FL)”.

---

> > ### Comment · Reviewer_ScDv · 2026-03-17
> >
> > I thank the authors for their thoughtful response to my comments and for their manuscript updates. The current draft addresses many of the concerns I raised in my original review. I have two points I want to revisit before submitting my recommendation:
> >
> > 1. **On PCA.** I understand that the authors removed the reference to PCA as their position that FL-Sailer is not an analysis of PCA failure. However, removing the mention of PCA while motivating their work with biologically relevant regions raises the question of why simple methods fail to do so. If the authors could include justification (either empirically through experimentation, theoretically, or by a reference to prior work), it would strengthen their manuscript.
> > 2. **Code.** I am generally not a fan of promissory notes for code release, particularly when the main contribution of the work is a framework. Please ensure that the repository will be complete (e.g. include experiment run files etc) that reflects the rigor and strength of the authors' manuscript.
> >
> > Thank you again for your thoughtful responses and updates.

---

> > > ### Author Response · Authors · 2026-03-18
> > > **Response**
> > >
> > > We really appreciate reviewer ScDv's effort and time. Thank you for bringing us valuable suggestions.
> > >
> > > A theoretical reason is that standard PCA and our sampling module solve different problems. PCA is a low-rank projection method: it returns latent directions, i.e., linear combinations of the original features, rather than a subset of actual genomic regions.
> > >
> > > In contrast, leverage-score column sampling is a column subset selection procedure: it selects actual columns of the original matrix while approximately preserving the dominant low-rank structure.
> > > Therefore, our point is not that PCA is uninformative, but that it does not directly produce a set of interpretable genomic regions, whereas column sampling does.
> > >
> > >
> > > This projection-versus-selection distinction is standard in the matrix approximation literature (e.g., Mahoney and Drineas, 2009; Drineas et al., 2008).
> > >
> > > We wish this could be helpful for your question.
> > >
> > >
> > > We have prepared an anonymous repository containing the FL-Sailer codebase, training scripts, and evaluation pipelines. We include the PBMC dataset configuration as a fully reproducible example, as this dataset is publicly available from 10x Genomics.
> > >
> > > The open-source code is available at:
> > > https://anonymous.4open.science/r/fl_sailer-A8D1
> > >
> > >
> > >
> > > References
> > >
> > > [1] M. W. Mahoney and P. Drineas. CUR matrix decompositions for improved data analysis. *PNAS*, 106(3):697–702, 2009.
> > > [2] P. Drineas, M. W. Mahoney, and S. Muthukrishnan. Relative-error CUR matrix decompositions. *SIAM J. Matrix Anal. Appl.*, 30(2):844–881, 2008.

---

> > > > ### Comment · Reviewer_ScDv · 2026-03-19
> > > >
> > > > I would like to thank the authors for their detailed response and engagement through the review process. I have no further questions or comments.

---

### Review · Reviewer_HhF1 · 2026-03-01

**Summary Of Contributions:**

This paper proposes FL-Sailer, a federated learning framework tailored for ultra-high-dimensional and highly sparse scATAC-seq data. The authors identify three main barriers to applying standard federated learning in this domain: excessive communication costs due to feature dimensionality, instability caused by sparsity, and severe cross-institutional batch effects. To address these issues, they introduce (1) adaptive leverage score sampling to significantly reduce dimensionality while preserving optimization geometry, and (2) an invariant VAE architecture that minimizes mutual information between latent representations and technical confounders. The paper also presents a convergence analysis that decomposes optimization error and sampling approximation error. Experiments on simulated and large real-world datasets demonstrate substantial communication reduction and competitive performance. Strengths include the relevance of the problem and the ambitious theoretical development, while weaknesses include strong theoretical assumptions and limited statistical validation in experiments.

**Additional Comments:**

Overall, this is an ambitious and technically detailed contribution that meaningfully advances federated learning for high-dimensional biological data. With stronger statistical validation and clearer discussion of theoretical assumptions, the work could provide a valuable contribution to both federated learning research and computational genomics applications.

**Audience:**

Yes

**Audience Explanation:**

The work connects federated learning, high-dimensional optimization, representation learning, and computational genomics, all of which are relevant to TMLR’s audience. In particular, the combination of communication-efficient FL design and theoretical convergence analysis in extreme-dimensional settings makes the findings broadly interesting to both methodological researchers and applied ML scientists.

**Broader Impact Concerns:**

The work promotes privacy-aware collaborative analysis of sensitive genomic data, thereby benefiting society. However, federated learning alone does not guarantee full protection against model inversion or gradient leakage attacks. The paper would benefit from briefly clarifying the scope and limits of its privacy guarantees.

**Claims And Evidence:**

Yes

**Claims Explanation:**

The paper provides substantial theoretical analysis and empirical evidence supporting its main claims. Communication efficiency is quantified, and convergence guarantees are derived using established federated optimization theory and subspace embedding results. However, claims of outperforming centralized training would benefit from stronger statistical validation, such as reporting standard deviations and significance tests. Additionally, some theoretical assumptions (e.g., non-critical gradient conditions) may be strong in practical deep learning settings.

**Requested Changes:**

To strengthen the paper, the authors should report the mean and standard deviation over multiple runs for all major experimental results and include statistical significance analysis. Clarification of strong theoretical assumptions and discussion of their practical validity would improve rigor. Additional ablation studies isolating the contributions of sampling and invariant representation learning would also clarify the source of performance gains. Runtime analysis and clearer hyperparameter tuning details would further enhance transparency.

---

> ### Author Response · Authors · 2026-03-12
> **Response**
>
> We thank Reviewer HhF1 for the thorough and constructive review.
> (1) Statistical validation. We now report mean ± standard deviation for all main tables to quantify variability in the revised manuscript.
>
> (2) Theoretical assumptions. We agree that these assumptions should be viewed as local/conditional rather than as globally valid descriptions of deep VAE training. Our theoretical goal is not to make stronger assumptions than the underlying FedAvg theory, but to show that, within the standard local analysis regime, leverage-score sampling preserves the geometry relevant to convergence and contributes only an explicit approximation term. We have revised the manuscript to make this scope clearer.
>
> (3) Ablation studies. We added a full ablation in Appendix F.4 (Table 7) that isolates the contributions of sampling and invariant representation learning. This confirms that the invariance objective is critical for handling cross-client batch effects.
>
> (4) Runtime and hyperparameters. We added a detailed wall-clock runtime breakdown in Appendix F.5 (Table 8), covering all sampling rates and datasets. You can refer to the response to Reviewer Vzvx as well.
>
> (5) Privacy scope. We have modified the privacy claims in the revised manuscript (Section 3.3) and clarified that our scope is limited to the standard cross-silo setting; we do not claim formal differential privacy or attack-specific guarantees in this work.

---

> > ### Comment · Reviewer_HhF1 · 2026-03-18
> > **Comment after Response**
> >
> > I appreciate the authors’ response. The primary concerns have been satisfactorily resolved.

---

### Decision · Action_Editor_j4ST · 2026-04-25

**Recommendation:** Accept as is

**Audience:**

Yes

**Audience Explanation:**

The reviewers were in consensus that the work is of interest to the TMLR audience - in particular those working on developing algorithms for analysis of genomics data sets.

**Claims And Evidence:**

Yes

**Claims Explanation:**

The reviewers conducted the thorough review of the manuscript and after revisions to align the claims with the evidence were in consensus that the claims are well-supported. The reviewers provided thoughtful feedback and the authors addressed the issues raised. The discussion of the manuscript improved it.